# VScan: Rethinking Visual Token Reduction for Efficient Large Vision-Language Models

**Ce Zhang[1],*    Kaixin Ma[2]    Tianqing Fang[2]    Wenhao Yu[2]    Hongming Zhang[2]
Zhisong Zhang[2]    Haitao Mi[2]    Dong Yu[2]**

[1]Carnegie Mellon University    [2]Tencent AI Lab

*Work done during an internship at Tencent AI Lab. Contact: `cezhang@cs.cmu.edu`.

**Reviewed on OpenReview:** `https://openreview.net/forum?id=KZYhyilFnt`

## Abstract

Recent Large Vision-Language Models (LVLMs) have advanced multi-modal understanding by incorporating finer-grained visual perception and encoding. However, such methods incur significant computational costs due to longer visual token sequences, posing challenges for real-time deployment. To mitigate this, prior studies have explored pruning unimportant visual tokens either at the output layer of the visual encoder or at the early layers of the language model. In this work, we revisit these design choices and reassess their effectiveness through comprehensive empirical studies of how visual tokens are processed throughout the visual encoding and language decoding stages. Guided by these insights, we propose VScan, a two-stage visual token reduction framework that addresses token redundancy by: (1) integrating complementary global and local scans with token merging during visual encoding, and (2) introducing pruning at intermediate layers of the language model. Extensive experimental results across four LVLMs validate the effectiveness of VScan in accelerating inference and demonstrate its superior performance over current state-of-the-arts on sixteen benchmarks. Notably, when applied to LLaVA-NeXT-7B, VScan achieves a $2.91\times$ speedup in prefilling and a $10\times$ reduction in FLOPs, while retaining 95.4% of the original performance. Code is available at `https://github.com/Tencent/SelfEvolvingAgent/tree/main/VScan`.

## 1 Introduction

Large Vision-Language Models (LVLMs) have emerged as a transformative advancement in multi-modal learning, achieving remarkable proficiency across a broad range of vision-language tasks [38, 34, 33, 56]. Recent advances in LVLMs [40, 32, 28, 5, 36, 45, 44] further enhance their capacity to process high-resolution images and multi-image/video inputs, enabling fine-grained perception in tasks such as video question answering [19, 61, 54], multi-image understanding [21, 29], and referential grounding [30, 46]. However, processing such rich visual inputs necessitates a substantial increase in the number of visual tokens, which often far exceeds the number of text tokens [36, 32]. For instance, LLaVA-NeXT [40] encodes up to 2,880 visual tokens for high-resolution images, while Qwen-2.5-VL [5] can process up to 16,384 tokens for multi-image or video inputs—orders of magnitude higher than typical text-only sequences. This leads to significantly longer input sequences and, due to the quadratic complexity of self-attention [58], incurs substantial computational and memory overhead, thereby limiting the scalability and real-time deployment of LVLMs in practical applications [12, 64].

Recognizing that not all visual tokens contribute meaningfully to the final LVLM response, recent works [12, 62, 70] have proposed visual token reduction techniques aimed at improving computational efficiency by pruning visually redundant or textually irrelevant tokens. These methods generally fall into two categories: (1) *Text-agnostic pruning approaches* [64, 59, 60] (Figure 1(a)), which prune visually redundant tokens based on their significance and uniqueness during the visual encoding stage, typically leveraging self-attention or [CLS] attention from the output layer of the visual encoder; and (2) *Text-aware pruning approaches* [71, 62, 65] (Figure 1(b)), which selectively remove tokens with low relevance to the text query during the early layers of language

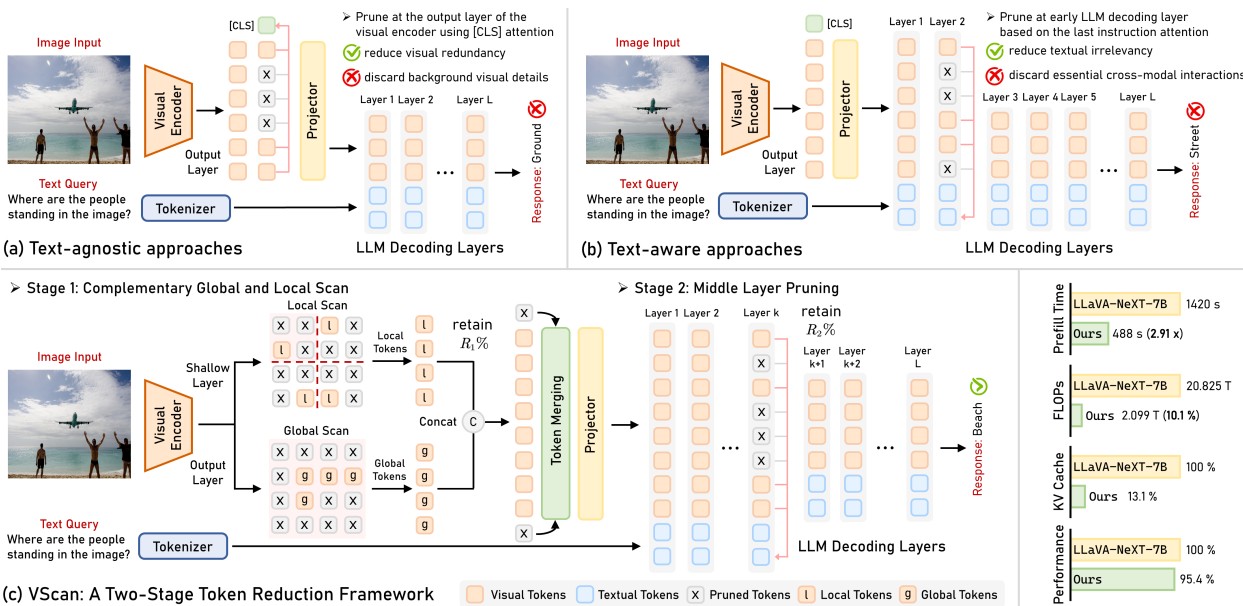

Figure 1: **Comparison of our VScan with representative text-agnostic approaches (*e.g.*, VisionZip [64]) and text-aware approaches (*e.g.*, FastV [12]).** In this work, we introduce VScan, a two-stage, training-free visual token reduction framework that can be seamlessly applied to various open-sourced LVLM architectures, delivering significant acceleration in inference with minimal performance loss.

decoding stage to preserve task-specific information while reducing computation. While these approaches have shown promising results, their performance is often constrained by their single-stage design and the lack of a systematic understanding of how visual tokens are processed and utilized throughout the *entire LVLM pipeline*.

In this work, we conduct an in-depth empirical analysis to reassess the effectiveness of these two prevailing pruning paradigms and distill insights that guide the design of more effective visual token reduction methods. Our study reveals two key observations: (1) In the visual encoding stage, the visual encoder attends to locally significant tokens in the shallow layers, focusing on fine-grained local details, while at deeper layers, it gradually shift its focus to a highly condensed set of tokens that encapsulate broader global context; (2) In the LLM decoding stage, early layers exhibit strong positional bias toward visual tokens appearing later in the sequence, neglecting their semantic relevance; as the layers deepen, cross-modal interactions begin to emerge, and output token probabilities typically converge in the mid-to-late layers where visual information is more effectively integrated into the language stream.

Building on these insights, we introduce VScan, a two-stage, training-free visual token reduction framework that enhances the efficiency of LVLMs by progressively pruning uninformative tokens during both visual encoding and language decoding stages, as shown in Figure 1(c). In the visual encoding stage, VScan employs a complementary global-local scan strategy to retain semantically important and spatially diverse tokens, followed by token merging to preserve comprehensive visual information. In the LLM decoding stage, VScan introduces middle layer pruning to further eliminate visual tokens with low relevance to the text query, while maintaining essential cross-modal interactions to minimize disruption to final task performance. Notably, VScan can be seamlessly integrated into diverse open-sourced LVLM architectures and is fully compatible with FlashAttention [18, 17], making it both practical and broadly applicable to real-world applications.

We comprehensively evaluate the effectiveness of VScan on LLaVA-1.5 [39], LLaVA-NeXT [40], Qwen-2.5-VL [5], and Video-LLaVA [36] across sixteen image and video understanding benchmarks. Extensive experimental results demonstrate VScan's generalizable effectiveness across diverse LVLM architectures and LLM scales, highlighting its advantageous performance-efficiency trade-off. Specifically, VScan achieves a 1.77× speedup on LLaVA-1.5-7B and a 2.91× speedup on LLaVA-NeXT-7B during prefilling, while retaining 96.7% and 95.4% of the original performance, respectively.

The contributions of this work are summarized as follows:

- We conduct comprehensive analyses to reveal how visual knowledge evolves throughout the entire LVLM, offering insights to inform the design of more effective visual token reduction strategies.

- We introduce VScan, a two-stage training-free visual token reduction framework that progressively eliminates unimportant visual tokens to reduce both visual redundancy and textual irrelevance.

- Extensive evaluations across sixteen benchmarks demonstrate that VScan consistently outperforms state-of-the-art methods in maintaining robust performance under constrained token budgets.

## 2 Related Work

**Efficient Large Vision-Language Models**. Building on powerful auto-regressive LLMs [57, 15], recent LVLMs typically adopt an encoder-projector-decoder architecture, where visual inputs are encoded into tokens and jointly processed with language sequences [38, 36, 4, 13, 56, 11]. However, as image resolution increases or the input scales to multi-image/video, the number of visual tokens grows proportionally, leading to a quadratic increase in computation cost and runtime due to the self-attention mechanism [58, 7, 14, 67], which limits the scalability of LVLMs in real-world applications [6, 12, 49, 31, 69, 68, 9]. To mitigate this issue, several LVLMs introduced specialized modules to enhance efficiency—such as the Q-Former in InstructBLIP [16] and the perceiver resampler [26] in OpenFlamingo [1]—that distill dense visual inputs into a compact set of features before LLM decoding. Orthogonal to these architectural strategies, FlashAttention [18, 17] has emerged as a widely adopted, hardware-aware optimization that accelerates attention computation by minimizing redundant memory access, offering substantial speedups without compromising performance.

**Vision Token Reduction in LVLMs**. Another line of work aims to improve model efficiency on the sequence dimension—pioneering works such as ToMe [6] and FastV [12] have explored strategies like visual token merging and text-guided pruning to improve the efficiency of LVLMs. Building on these advances, subsequent approaches can be broadly divided into two main categories: (1) *Text-agnostic pruning approaches* [52, 3, 70, 64, 59, 60], which identify and remove redundant or uninformative visual tokens during the visual encoding stage. For instance, VisionZip [64] selects dominant tokens based on [CLS] attention scores, while FOLDER [59] introduces token merging with reduction overflow in the final blocks of the visual encoder. (2) *Text-aware pruning approaches* [71, 62, 41, 65, 55], which aim to remove visual tokens that are irrelevant to the text query during the LLM decoding stage. For instance, SparseVLM [71] proposes an iterative sparsification strategy that selects visual-relevant text tokens to rate the significance of vision tokens, and PyramidDrop [62] performs progressive pruning at multiple decoding layers to balance efficiency and context preservation. In this work, we present a comprehensive analysis of how LVLMs process visual tokens during both the visual encoding and language decoding stages, and propose a corresponding two-stage approach, VScan, to effectively improve the inference efficiency of LVLMs while maintaining robust performance.

## 3 Empirical Analysis

In this section, we provide a comprehensive analysis of how LVLMs process visual tokens during both the visual encoding and language decoding stages, offering empirical guidance for designing more effective visual token reduction strategies.

**Preliminary: Architecture of LVLMs**. We consider an LVLM parameterized by $\theta$, which consists of three major components: a visual encoder, a feature projector, and an LLM decoder. Given an image input, the visual encoder processes the image patches, and the projector converts them into $n$ visual tokens $\mathbf{x}_V = \{x_V^i\}_{i=1}^n$. These visual tokens are then concatenated with the tokenized textual query $\mathbf{x}_T$ and fed into the LLM decoder for auto-regressive next-token generation, represented as $y_t \sim p_\theta(y_t|\mathbf{x}_V, \mathbf{x}_T, \mathbf{y}_{<t})$, where the next token $y_t$ is sampled from the output probability distribution $p_\theta(\cdot)$, and $\mathbf{y}_{<t}$ denotes the sequence of tokens generated prior to timestep $t$.

**Rethinking Visual Redundancy Reduction**. To address visual redundancy in token representations, recent studies [70, 64] have proposed *text-agnostic approaches* that retain visual tokens with high [CLS] attention at the output layer of the ViT-based visual encoder. While effective to some extent, this strategy raises an important question: *Is relying solely on output [CLS] attention truly sufficient to capture all*

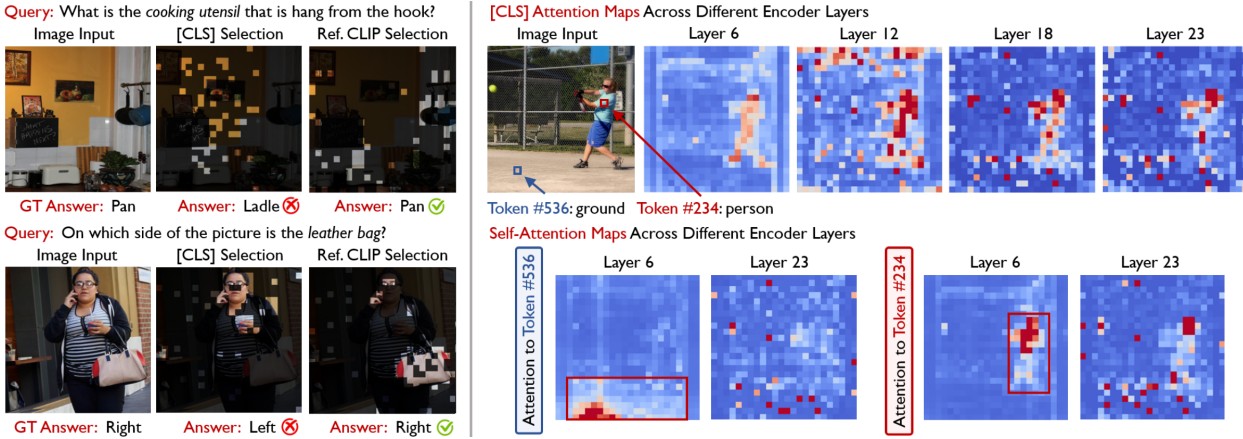

Figure 2: **Empirical study on visual redundancy reduction**. (*Left*) We illustrate two failure cases where relying solely on the output [CLS] attention leads to incorrect predictions. For comparison, we include reference token selections from CLIP-ViT-L-336px [51], following Gandelsman *et al.* [22], which highlight regions of interest relevant to the text query. (*Right*) We visualize the [CLS] attention maps and self-attention maps of representative tokens (*e.g.*, #536: *ground*, #234: *person*) across different encoding layers, illustrating how attention patterns evolve from localized focus in shallow layers to broader global context in deeper layers.

*task-relevant visual information?* Upon closer examination, we identify a clear yet often overlooked limitation of these approaches: they tend to favor tokens corresponding to visually salient objects, while aggressively discarding background visual details that may carry essential semantic information. As illustrated by the examples in Figure 2 (*Left*), output [CLS] attention is incorrectly directed to the *wall* and *person*, ignoring the actual targets—the *pan* and *leather bag*—leading to incorrect model responses.

To better understand and overcome this limitation, we analyze how visual information is processed across the visual encoding layers in LVLMs. Specifically, we visualize both the [CLS] attention and self-attention of representative tokens across different visual encoding layers, as illustrated in Figure 2 (*Right*). Our observations are as follows: (1) In the shallow layers, the [CLS] attention maps capture fine-grained local details across the image. In contrast, in the deeper layers, the attention becomes increasingly concentrated on the main entities, reflecting their global semantic relevance; (2) The self-attention maps for representative visual tokens reveal a similar local-to-global trend: in the shallow layers, these tokens primarily attend to nearby regions with similar semantic meaning, while in the deeper layers, their attention becomes more dispersed, integrating context from the entire image. These findings highlight a gradual transition in the visual encoder from capturing low-level local details to modeling high-level, globally relevant semantics, suggesting that relying solely on the output layer may overlook the rich local information encoded in the shallow layers.

**Rethinking Textual Irrelevance Reduction**. While prior studies [12, 71, 37] have proposed effective *text-aware approaches* for pruning visual tokens at early layers during LLM decoding, a critical question remains: *Are early layers the optimal stage for pruning visual tokens to minimize their impact on the model's final response?* To investigate this, we conduct three empirical studies on POPE [35] and GQA [25], analyzing how the model's knowledge and predictions evolve during the decoding process:

- **Study 1**: *How does position bias in token selection evolve across LLM layers?* Specifically, we visualize the distribution of the retained tokens selected by the attention score of the last instruction token [12] across LLM layers using LLaVA-1.5-7B. As shown in Figure 3 (*Left*), early layers (*e.g.*, layers 2 and 8) tend to select tokens at the bottom of the image, reflecting an *inherent LLM position bias*, as the last instruction token primarily attends to nearby tokens and focuses on local context [58, 50], and flattened visual tokens from the bottom of the image are positioned closest to the instruction tokens in the sequence. As the LLM layers deepen, this undesirable position bias diminishes and the focus shifts toward the center of the image, which is more intuitive since the center of the image typically carries the most informative and task-relevant features [2, 8].

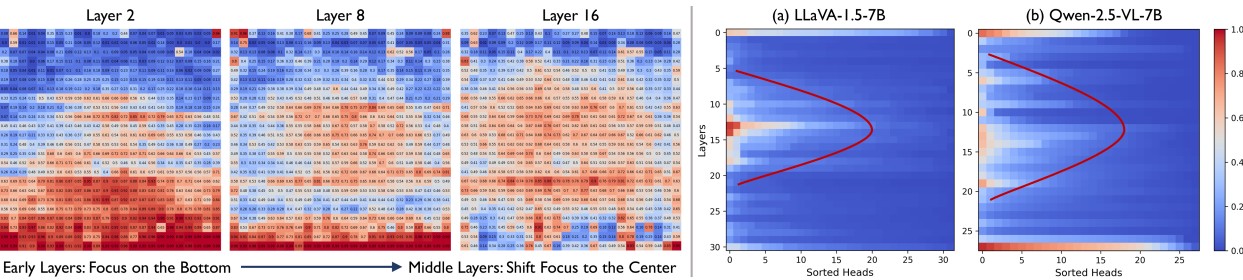

Figure 3: (*Left*) **Study 1**: Distribution of retained tokens at a 50% reduction rate in layers 2, 8, and 16 of LLaVA-1.5-7B on POPE [35]; (*Right*) **Study 2**: Sum of visual attention across different attention heads and LLM layers using LLaVA-1.5-7B and Qwen-2.5-VL-7B on POPE [35].

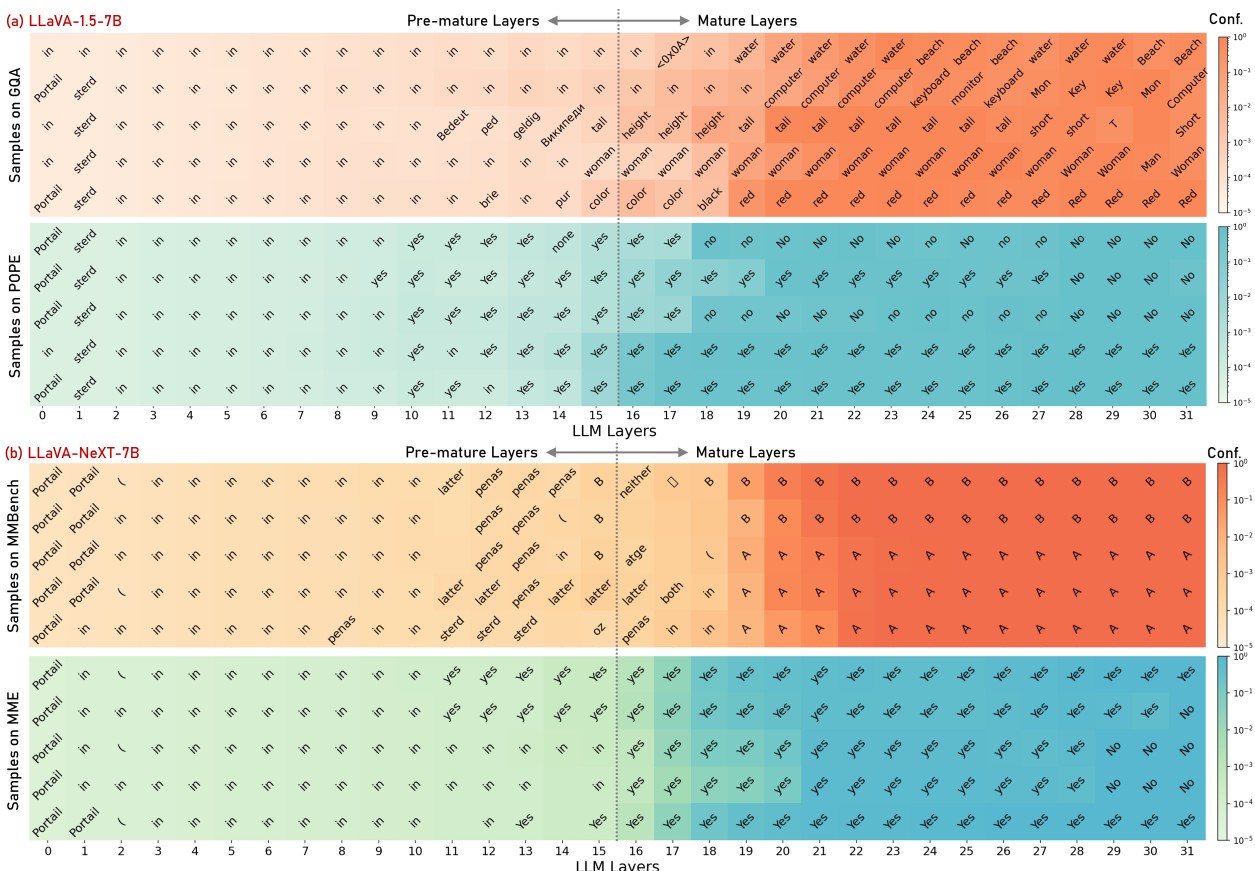

Figure 4: **Study 3**: Visualization of next-token predictions derived from the output hidden states of each LLM layer using (a) LLaVA-1.5-7B; (b) LLaVA-NeXT-7B. Darker colors indicate higher prediction confidence.

- **Study 2**: *From which layer does the LLM begin to gather and process visual information?* We visualize the sum of attention received by all visual tokens from the last instruction token across different LLM layers using LLaVA-1.5-7B and Qwen-2.5-VL-7B in Figure 3 (*Right*). The red curve in each plot highlights the layer-wise attention patterns directed towards visual information. We observe that the middle LLM layers are primarily responsible for interacting with the visual tokens, whereas the early and deep layers focus predominantly on processing textual information.

- **Study 3**: *At which LLM layer do next-token predictions begin to converge?* In Figure 4, we provide an interpretation of the hidden states across different LLM layers using LLaVA-1.5-7B and LLaVA-NeXT-7B. Specifically, we feed the hidden states from each LLM decoding layer into the final linear projection layer to obtain vocabulary logits and intermediate next-token predictions. We observe that in more challenging

open-ended tasks like GQA, the next-token predictions stabilize around LLM layer 20, whereas in simpler yes/no tasks such as POPE, the predictions converge earlier, around LLM layer 16. Our findings indicate that early layers are still forming core cross-modal semantics, and pruning them risks disrupting essential grounding. In contrast, by the middle layers, next-token predictions have largely stabilized, meaning that these layers contribute diminishing semantic change. This directly motivates pruning in the middle-to-late layers rather than the early layers.

These findings collectively suggest that early layers are suboptimal for pruning due to position bias and limited engagement with visual content. In contrast, pruning at middle layers is more appropriate as it better preserves critical cross-modal interactions and minimizes disruption to model predictions.

## 4 Method

We introduce VScan, a training-free approach that progressively prunes uninformative tokens in both visual encoding and LLM decoding stages to accelerate LVLM inference, as illustrated in Figure 1(c).

### 4.1 Reducing Visual Redundancy via Complementary Global and Local Scans

Motivated by the observations in Section 3, we design two complementary token selection schemes for the visual encoding stage, namely global and local scan, which select important tokens based on both local and global significance, enabling the capture of more comprehensive visual details.

**Global Scan**. Given that the final layers of visual encoders capture global information, we follow recent works [70, 64] to select global tokens that receive the most attention from the [CLS] token $x_{[\texttt{CLS}]}$ in the output layer (*e.g.*, the penultimate layer in LLaVA-1.5 [39]). Specifically, the [CLS] attention computation for each attention head can be represented by

$$Q_{[\texttt{CLS}]} = x_{[\texttt{CLS}]} W_Q^h, \quad K_V = \mathbf{x}_V W_K^h, \quad S_{[\texttt{CLS}]}^h = \texttt{Softmax}\left(\frac{Q_{[\texttt{CLS}]} K_V^\top}{\sqrt{D}}\right), \tag{1}$$

where $W_Q^h$ and $W_K^h$ represent the projections weights for head $h \in [1, H]$, $D$ denotes the hidden state size, and $S_{[\texttt{CLS}]}^h$ represents the [CLS] attention. The global tokens are then selected by

$$\mathbf{x}_V^{\texttt{g}} = \left\{ x_V^i \in \mathbf{x}_V \;\middle|\; S_{[\texttt{CLS}]}^{\texttt{avg}} \geq \tau \right\}, \text{ where } S_{[\texttt{CLS}]}^{\texttt{avg}} = \frac{1}{H}\sum_{h=1}^{H} S_{[\texttt{CLS}]}^h. \tag{2}$$

Here, $\tau$ is a soft threshold based on a top percentile of attention scores, set to retain a target number of tokens. Note that for LVLMs without a [CLS] token (*e.g.*, Qwen-2.5-VL [5]), we can similarly select the tokens using self-attention, *i.e.*, the average attention each visual token receives from others.

**Local Scan**. To complement the global tokens and capture finer local details, we divide the image into non-overlapping windows and select the locally important tokens with the highest [CLS] attention from the shallow layer $l$ within each window. Specifically, we allocate token budgets uniformly across windows, and select local tokens from each window as:

$$\mathbf{x}_V^{\texttt{l}} = \bigcup_{w=1}^{W} \left\{ x_V^j \in \mathbf{x}_V^w \;\middle|\; S_{[\texttt{CLS}]}^{\texttt{avg}} \geq \tau_w \right\} \tag{3}$$

where $w$ denotes the window index, $\mathbf{x}_V^w$ represents the set of all tokens within the window, and $\tau_w$ is the soft threshold for window $w$. The final set of selected tokens is the union of global and local tokens, $\mathbf{x}_V^{\texttt{selected}} = \mathbf{x}_V^{\texttt{g}} \cup \mathbf{x}_V^{\texttt{l}}$, resulting in a retention rate of $R_1\%$. By default, we balance the selection such that $|\mathbf{x}_V^{\texttt{g}}| = |\mathbf{x}_V^{\texttt{l}}|$, *i.e.*, half of the retained tokens are global and half are local.[1]

**Token Merging**. To alleviate information loss, we introduce a similarity-based token merging strategy that merges unselected visual tokens with their most similar selected counterparts. Specifically, for each unselected

---

[1]To avoid duplication, we prioritize local tokens and exclude them from global token selection. Empirically, we find that this prioritization does not significantly affect the final performance.

token $x_V^u$, we identify its most similar selected token $x_V^s \in \mathbf{x}_V^{\texttt{selected}}$ based on the highest cosine similarity. Once all unselected tokens are assigned to their closest selected tokens, we apply average merging [6] within each group to obtain the final merged representation $\mathbf{x}_V^{\texttt{merged}}$. Specifically, for each selected token $\mathbf{x}_V^s$, we compute the average token representation by

$$\mathbf{x}_V^{\texttt{merged}} = \frac{1}{|\mathcal{U}^s| + 1} \left( \sum_{u \in \mathcal{U}^s} x_V^u + \mathbf{x}_V^s \right), \tag{4}$$

where $\mathcal{U}^s$ denotes the set of unselected tokens associated with the selected token $\mathbf{x}_V^s$, and $|\mathcal{U}^s|$ indicates the cardinality of this set.

## 4.2 Reducing Textual Irrelevance via Middle Layer Pruning

After selecting visually significant tokens, we further refine the token set based on their relevance to the text query. Building on the empirical insights from Section 3, we design our approach to prune tokens at the mature middle layers of the LLM, aiming to avoid position bias, preserve cross-modal interactions, and minimize the impact on final predictions. Specifically, we compute the attention between all visual tokens and the last instruction token at middle layer $k$, denoted as

$$Q_T = x_T^{\texttt{last}} W_Q^h, \quad K_V = \mathbf{x}_V^{\texttt{merged}} W_K^h, \quad S_{\texttt{text}}^h = \texttt{Softmax}\left( \frac{Q_T K_V^\top}{\sqrt{D}} \right). \tag{5}$$

We similarly average the attention scores across different attention heads and select $R_2\%$ textually relevant tokens with the highest average text attention. This allows us to retain a set of visual tokens that are both visually significant and textually relevant, contributing the most to an accurate response.

**Empirical Validation**. We conduct a comparative analysis on the GQA benchmark using LLaVA-1.5-7B [39] to examine the effect of pruning tokens at different LLM layers, while keeping the average reduction rate consistent across settings. To better highlight the impact of pruning depth, we first apply a global scan to reduce the visual tokens to 288/144 (*i.e.*, $R_1 = 50\%/25\%$), and then perform pruning at various LLM layers to reach an average retention rate of 75% during the LLM decoding stage. As shown in Table 1, pruning at middle LLM layers (*e.g.*, layers 16 or 20) yields the best performance, whereas pruning at earlier layers (*e.g.*, layer 2) leads to up to a 1.9% drop in accuracy. These results align with our empirical insights in Section 3 and validate the effectiveness of pruning at middle layers to remove textual irrelevance in LVLMs.

Table 1: **Comparative study of pruning visual tokens at different LLM layers**. $R_1$ denotes the retention rate in the visual encoding stage, while $k$ and $R_2$ indicate the pruning layer and retention rate in the LLM decoding stage, respectively.

| Settings | $R_1 = 50\%$ | $R_1 = 25\%$ |
|---|---|---|
| $k = 2$, $R_2 = 73.3\%$ | 59.6 | 56.8 |
| $k = 8$, $R_2 = 66.7\%$ | 59.6 | 57.2 |
| $k = 12$, $R_2 = 60.0\%$ | 59.6 | 58.6 |
| $k = 16$, $R_2 = 50.0\%$ | **60.7** | **58.7** |
| $k = 20$, $R_2 = 33.3\%$ | 60.6 | **58.7** |
| $k = 24$, $R_2 = 0.0\%$ | 60.2 | 58.4 |

**Remarks on KV Cache and FlashAttention**. Our proposed VScan is fully compatible with standard KV caching mechanisms, as pruning occurs before visual tokens are added to the cache. Consequently, the KV cache stores fewer entries while its structure and format remain unchanged. VScan is also compatible with FlashAttention [17, 18], as we recompute the attention scores for the last instruction token using vanilla attention calculation [58] outside the standard LLM layers.

# 5 Experiments

In this section, we validate the effectiveness of our VScan on four widely used LVLMs, evaluating its performance across various benchmarks and comparing it with other state-of-the-art approaches.

## 5.1 Experimental Settings

**Models**. We evaluate the general effectiveness of VScan by applying it to four popular LVLMs with diverse architectures. Following prior work in this field, we first compare performance on LLaVA-1.5-7B [39], a widely adopted academic baseline that maps each image input to 576 tokens, and LLaVA-NeXT-7B [40],

which improves high-resolution understanding by encoding an image into up to 2,880 visual tokens. We further assess Video-LLaVA-7B [36], which extends the LLaVA framework to videos, processing up to 8 frames with 2,048 visual tokens. Finally, we are among the first to report experimental results on the recent Qwen-2.5-VL [5], tested across different LLM sizes (3B, 7B, 32B). This model incorporates dynamic resolution processing to handle images of varying sizes, supporting visual token counts ranging from 4 to 16,384.

**Benchmarks and Metrics**. We conduct extensive experiments on 9 standard image understanding benchmarks, including visual question answering benchmarks such as GQA [25], ScienceQA [43], VQAv2 [23], TextVQA [53] and VizWiz [24]; multi-modal reasoning benchmarks such as MMBench [42], MMBench-CN [42], MME [20], and POPE [35]. We also include evaluations on 3 more challenging referring grounding tasks using RefCOCO [30], RefCOCO+ [30], and RefCOCOg [46], and report the accuracy achieved by different approaches. In these grounding tasks, a localization is considered correct if the predicted bounding box has an IoU score of at least 0.5 with the ground truth. Additionally, we evaluate our approach on 4 video question answering benchmarks: TGIF [27], MSVD [10], MSRVTT [63], and ActivityNet [66]. We follow previous work [45, 62] to utilize both accuracy and the ChatGPT score[2] as key performance metrics for these video-based benchmarks. The evaluation protocol and prompts are detailed in Section B.4.

**Baselines**. We compare the performance of our approach with 6 state-of-the-art visual token pruning methods: ToMe [6], FastV [12], SparseVLM [71], HiRED [3], PyramidDrop [62], and VisionZip [64]. (1) ToMe [6], which uses bipartite soft matching to iteratively merge similar tokens within ViT layers; (2) FastV [12], which drops visual tokens in early layers of the LLM, guided by text-oriented attention score; (3) SparseVLM [71], which selects vision-relevant text tokens to evaluate the significance of visual tokens; (4) HiRED [3], which dynamically assigns token budgets to sub-images for high-resolution image inputs; (5) PyramidDrop [62], which divides the LLM into stages and drops a portion of visual tokens at the end of each stage; (6) VisionZip [64], which selects a set of dominant tokens and merges unselected tokens contextually. To ensure a fair comparison, we directly report the results of these baselines from their respective original papers unless stated otherwise.

**Implementation Details**. We adhere to the default inference settings for each evaluated LVLM as specified in their respective codebases. Additionally, we perform local scan at a shallow layer, specifically at $l = 6$ for LLaVA-series models and $l = 8$ for Qwen-2.5-VL. For LLM-stage pruning, we select the middle layer as $k = 16$ for LLaVA-series models and $k = 14$ for Qwen-2.5-VL-7B. For Qwen-2.5-VL-3B and -32B, we similarly select the exact middle layer for pruning. By default, we fix the retention rate at the LLM middle layer to $R_2 = 33.3\%$, and adjust $R_1$ accordingly to achieve the target average reduction rate. For instance, to achieve an average retention rate of 11.1%, we set $R_1 = 16.7\%$ and $R_2 = 33.3\%$. Note that these design choices are grounded in our analysis in Section 3, and we also provide comprehensive ablation studies in Section 5.3. Code will be made publicly available upon acceptance to facilitate reproducibility.

## 5.2 Results and Discussions

**Results on LLaVA-1.5**. In Table 2, we apply our approach to LLaVA-1.5-7B [39] and compare its performance with other baselines across 9 image understanding tasks. Following previous work [71, 64] in this field, we present a comparative analysis of performance across three settings, retaining an average of 192, 128, and 64 visual tokens across all 32 layers in LLaMA [57], corresponding to reduction rates of 66.7%, 77.8%, and 88.9%, respectively. We also calculate and present the percentage of performance retention in the final column, using the vanilla LLaVA-1.5 performance as the 100% upper limit. As shown in the table, with only 128 and 192 tokens per image instead of the original 576, our approach nearly retains the performance of the original LLaVA-1.5, with only negligible performance declines of 1.0% and 1.2%, respectively. Our approach becomes even more advantageous with higher reduction rates: With an aggressive 88.9% reduction rate, our approach results in only a 3.3% degradation in average performance across benchmarks, outperforming the second-best VisionZip [64] by a substantial margin of 4.0%. These results suggest that our method effectively maintains high performance while reducing the number of visual tokens processed.

**Results on LLaVA-NeXT**. In Table 3, we further compare the performance achieved by our approach with other state-of-the-arts on the more advanced LLaVA-NeXT-7B [40]. We present the performance of all approaches under a fixed budget of 320 tokens per image, corresponding to an 88.9% reduction rate. Our

---

[2]Evaluated using *gpt-3.5-turbo*: https://platform.openai.com/docs/models/gpt-3.5-turbo.

Table 2: **Performance comparisons on LLaVA-1.5-7B [39] across 9 image understanding benchmarks**. The best results in each setting are **bolded**, and the second-best are underlined.

| Method | Venue | GQA | MMB | MMB$^{CN}$ | MME | POPE | SQA$^{IMG}$ | VQA$^{V2}$ | VQA$^{Text}$ | VizWiz | Average |
|---|---|---|---|---|---|---|---|---|---|---|---|
| *Upper Bound, 576 Tokens (100%), 3.817 TFLOPs* | | | | | | | | | | | |
| LLaVA-1.5-7B [39] | *CVPR'24* | 61.9 | 64.7 | 58.1 | 1862 | 85.9 | 69.5 | 78.5 | 58.2 | 50.0 | 100.0% |
| *Retain 192 Tokens in Average (↓ 66.7%), ~1.253 TFLOPs* | | | | | | | | | | | |
| ToMe [6] | *ICLR'23* | 54.3 | 60.5 | - | 1563 | 72.4 | 65.2 | 68.0 | 52.1 | - | 88.5% |
| FastV [12] | *ECCV'24* | 52.7 | 61.2 | 57.0 | 1612 | 64.8 | 67.3 | 67.1 | 52.5 | 50.8 | 90.4% |
| SparseVLM [71] | *ICML'25* | 59.5 | **64.1** | 53.7 | 1787 | 85.3 | 68.7 | 75.6 | **57.8** | 50.5 | 97.6% |
| PyramidDrop [62] | *CVPR'25* | 57.3 | 63.3 | 56.8 | 1797 | 82.3 | **69.0** | 75.1 | 56.5 | **51.1** | 97.2% |
| VisionZip [64] | *CVPR'25* | 59.3 | 63.0 | - | 1783 | 85.3 | 68.9 | 77.4 | 57.3 | - | 97.8% |
| **VScan (Ours)** | - | **60.6** | 63.9 | **57.4** | **1806** | **86.2** | 68.6 | **77.8** | 57.7 | 50.4 | **99.0%** |
| *Retain 128 Tokens in Average (↓ 77.8%), ~0.833 TFLOPs* | | | | | | | | | | | |
| ToMe [6] | *ICLR'23* | 52.4 | 53.3 | - | 1343 | 62.8 | 59.6 | 63.0 | 49.1 | - | 80.4% |
| FastV [12] | *ECCV'24* | 49.6 | 56.1 | 56.4 | 1490 | 59.6 | 60.2 | 61.8 | 50.6 | 51.3 | 85.4% |
| SparseVLM [71] | *ICML'25* | 58.4 | **64.5** | 51.1 | 1746 | 85.0 | 68.6 | 73.8 | 56.7 | 51.4 | 96.4% |
| PyramidDrop [62] | *CVPR'25* | 57.1 | 61.6 | 56.6 | 1761 | 82.3 | 68.4 | 72.9 | 56.6 | 51.0 | 96.2% |
| VisionZip [64] | *CVPR'25* | 57.6 | 62.0 | - | 1763 | 83.2 | **68.9** | 75.6 | 56.8 | - | 96.2% |
| **VScan (Ours)** | - | **59.8** | 63.0 | **58.0** | **1792** | **86.1** | 68.9 | **77.1** | 57.3 | **51.7** | **98.8%** |
| *Retain 64 Tokens in Average (↓ 88.9%), ~0.415 TFLOPs* | | | | | | | | | | | |
| ToMe [6] | *ICLR'23* | 48.6 | 43.7 | - | 1138 | 52.5 | 50.0 | 57.1 | 45.3 | - | 70.1% |
| FastV [12] | *ECCV'24* | 46.1 | 48.0 | 52.7 | 1256 | 48.0 | 51.1 | 55.0 | 47.8 | 50.8 | 76.7% |
| SparseVLM [71] | *ICML'25* | 53.8 | 60.1 | 46.1 | 1589 | 77.5 | **69.8** | 68.2 | 53.4 | 50.1 | 90.4% |
| PyramidDrop [62] | *CVPR'25* | 47.5 | 58.8 | 50.5 | 1561 | 55.9 | 69.2 | 69.2 | 50.6 | 50.7 | 86.6% |
| VisionZip [64] | *CVPR'25* | 55.1 | 60.1 | - | 1690 | 77.0 | 69.0 | 72.4 | 55.5 | - | 92.7% |
| **VScan (Ours)** | - | **58.3** | **62.1** | **55.7** | **1698** | **85.0** | 69.1 | **75.4** | **55.6** | **51.8** | **96.7%** |

Table 3: **Performance comparisons on LLaVA-NeXT-7B [40] across 9 image understanding benchmarks**. The best results in each setting are **bolded**, and the second-best are underlined.

| Method | Venue | GQA | MMB | MMB$^{CN}$ | MME | POPE | SQA$^{IMG}$ | VQA$^{V2}$ | VQA$^{Text}$ | VizWiz | Average |
|---|---|---|---|---|---|---|---|---|---|---|---|
| *Upper Bound, 2,880 Tokens (100%), ~20.825 TFLOPs* | | | | | | | | | | | |
| LLaVA-NeXT-7B [40] | *CVPR'24* | 64.2 | 67.4 | 60.6 | 1851 | 86.5 | 70.1 | 81.8 | 61.3 | 57.6 | 100.0% |
| *Retain 320 Tokens in Average (↓ 88.9%), ~2.099 TFLOPs* | | | | | | | | | | | |
| FastV [12] | *ECCV'24* | 55.9 | 61.6 | 51.9 | 1661 | 71.7 | 62.8 | 72.9 | 55.7 | 53.1 | 88.7% |
| HiRED [3] | *AAAI'25* | 59.3 | 64.2 | 55.9 | 1690 | 83.3 | 66.7 | 75.7 | 58.8 | **54.2** | 93.9% |
| PyramidDrop [62] | *CVPR'25* | 56.4 | 63.4 | 56.2 | 1663 | 77.6 | **67.5** | 73.5 | 54.4 | 54.1 | 91.4% |
| VisionZip [64] | *CVPR'25* | 59.3 | 63.1 | - | 1702 | 82.1 | 67.3 | 76.2 | **58.9** | - | 94.0% |
| **VScan (Ours)** | - | **60.7** | **65.3** | **57.8** | **1767** | **85.1** | 66.9 | **77.1** | 58.0 | 53.8 | **95.4%** |

approach continues to achieve superior performance on LLaVA-NeXT-7B [40], attaining the best performance on 6 out of 9 benchmarks, and achieving 95.4% of the original LLaVA-NeXT-7B [40] performance with only 11.1% of the token budget without additional training.

**Results on Qwen-2.5-VL**. To further validate the general effectiveness of our approach, we apply it to the more advanced Qwen-2.5-VL [5] with three different LLM scales and visualize the performance on three general visual understanding benchmarks in Figure 5. The full numerical results are also provided in Appendix A.7. As shown, our approach consistently outperforms FastV [12], PDrop [62] and VisionZip [64] across all retention rates (from 11.1% to 50%) and model scales. We also observe that on MME, both FastV and PDrop exhibit degraded performance under low token budgets as the LLM scales from 7B to 32B. We speculate that this is due to larger LLMs introducing stronger language priors, which can hinder the accurate

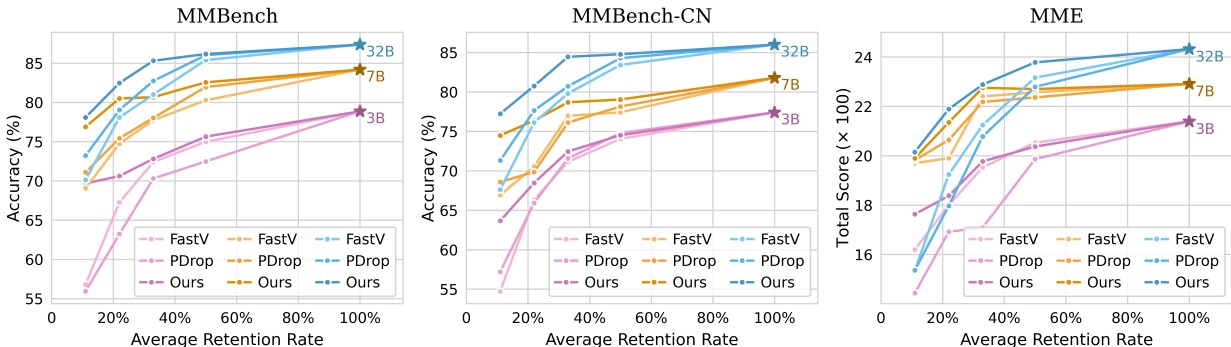

Figure 5: **Performance comparisons on Qwen-2.5-VL [5] with different LLM sizes (3B/7B/32B) across 3 image understanding benchmarks**. We present the performance of different approaches at 4 various retention rates, along with the original model performance without token reduction.

Table 4: **Performance comparisons on Qwen-2.5-VL-7B [5] across 3 referring expression comprehension benchmarks: RefCOCO, RefCOCO+, and RefCOCOg**. The best results in each setting are **bolded**, and the second-best are underlined. †Evaluation is based on our re-implementation.

| Method | Venue | RefCOCO | | | RefCOCO+ | | | RefCOCOg | | Average |
|---|---|---|---|---|---|---|---|---|---|---|
| | | val | testA | testB | val | testA | testB | val | test | |
| *Upper Bound, 4~16384 Tokens (100%)* | | | | | | | | | | |
| Qwen-2.5-VL-7B [5] | *arXiv'25* | 89.45 | 92.56 | 85.16 | 83.50 | 89.02 | 79.15 | 86.76 | 87.24 | 100.0% |
| *Retain 75% Tokens in Average (↓ 25%)* | | | | | | | | | | |
| FastV† [12] | *ECCV'24* | 85.27 | 87.84 | 82.28 | 79.02 | 82.95 | 72.86 | 82.95 | 83.32 | 94.8% |
| PyramidDrop† [62] | *CVPR'25* | 87.79 | 91.00 | 83.22 | 81.48 | 86.55 | 74.02 | 84.62 | 85.10 | 97.2% |
| VisionZip† [64] | *CVPR'25* | 88.33 | 91.55 | 83.68 | 81.99 | 87.28 | 74.09 | 85.08 | 85.92 | 97.9% |
| **VScan (Ours)** | - | **88.75** | **91.94** | **83.96** | **82.39** | **87.90** | **74.15** | **85.54** | **86.55** | **98.3%** |
| *Retain 50% Tokens in Average (↓ 50%)* | | | | | | | | | | |
| FastV† [12] | *ECCV'24* | 73.85 | 73.38 | 74.21 | 66.75 | 68.88 | 62.65 | 71.06 | 71.86 | 81.2% |
| PyramidDrop† [62] | *CVPR'25* | 77.52 | 80.82 | 72.07 | 70.27 | 75.48 | 63.33 | 74.86 | 75.65 | 85.1% |
| VisionZip† [64] | *CVPR'25* | 80.92 | 84.55 | 76.91 | 74.02 | 79.52 | 67.89 | 78.12 | 79.34 | 89.7% |
| **VScan (Ours)** | - | **86.78** | **90.74** | **82.37** | **79.99** | **86.12** | **71.67** | **84.03** | **84.44** | **96.1%** |
| *Retain 25% Tokens in Average (↓ 75%)* | | | | | | | | | | |
| FastV† [12] | *ECCV'24* | 43.57 | 46.81 | 40.86 | 39.47 | 43.78 | 36.02 | 43.04 | 42.69 | 48.5% |
| PyramidDrop† [62] | *CVPR'25* | 46.46 | 53.83 | 37.23 | 42.29 | 47.76 | 32.81 | 45.32 | 44.91 | 50.4% |
| VisionZip† [64] | *CVPR'25* | 60.12 | 66.95 | 55.41 | 58.33 | 64.88 | 48.52 | 57.09 | 56.44 | 67.8% |
| **VScan (Ours)** | - | **74.32** | **79.05** | **68.22** | **67.22** | **73.72** | **58.95** | **69.42** | **69.43** | **80.7%** |

selection of critical visual tokens. In contrast, our two-stage reduction strategy mitigates this bias and remains robust across LLM scales, demonstrating superior performance.

We extend our comparisons to more challenging grounding tasks and present the performance of different approaches in Table 4. Compared to image understanding tasks, these grounding tasks require higher token budgets to preserve visual information necessary for precise localization. A 75% reduction rate, for instance, halves the performance of FastV [12] and PDrop [62]. In this challenging scenario, our approach still robustly maintains 80.7% of the original performance. Additionally, our approach achieves 96.1% of the original performance with only 50% of the visual tokens. Qualitative results in Figure A2 further demonstrate that our approach effectively retains the critical tokens for accurate localization in challenging cases.

**Results on Video-LLaVA**. Finally, we validate the effectiveness of our approach on video understanding tasks and compare its performance against other approaches on Video-LLaVA-7B [36], as shown in Table 5. Specifically, we report the accuracy and GPT-evaluated scores for each benchmark to assess the quality of the responses. The experimental results demonstrate that, even under a strict 25% token budget, our approach

| Method | TGIF | | MSVD | | MSRVTT | | ActivityNet | |
|---|---|---|---|---|---|---|---|---|
| | Acc. | Score | Acc. | Score | Acc. | Score | Acc. | Score |
| Video-LLaVA-7B [36] | 47.0 | 3.40 | 70.5 | 3.92 | 58.3 | 3.51 | 42.2 | 3.37 |
| FastV† [12] | 42.7 | 3.19 | 67.4 | 3.83 | 53.6 | 3.40 | 36.1 | 3.15 |
| PyramidDrop† [62] | 44.1 | 3.26 | 66.7 | 3.81 | 56.1 | 3.45 | 37.4 | 3.15 |
| VisionZip† [64] | 44.0 | 3.28 | 68.2 | 3.85 | 55.4 | 3.42 | 39.0 | 3.23 |
| **VScan (Ours)** | **46.9** | **3.35** | **69.8** | **3.93** | **57.1** | **3.48** | **42.6** | **3.34** |

Table 5: **Performance comparisons on Video-LLaVA-7B [36] across 4 video understanding tasks with a 75% reduction rate**. The best results are **bolded**, and the second-best are underlined. †Evaluation is based on our re-implementation.

Table 6: **Ablation experiment using LLaVA-1.5-7B with an average reduction rate of 11.1% on GQA and MME**. We ablate the effects of (a) retention rates $R_1$ and $R_2$; (b) the proportion of global and local tokens; and (c) encoding layer $l$ for the local scan, while keep all other settings fixed.

**(a) Retention rates $R_1$ and $R_2$.**

| $R_1$ | $R_2$ | GQA | MME |
|---|---|---|---|
| 11.1% | 100.0% | 56.7 | 1651 |
| 13.3% | 66.7% | 57.1 | 1683 |
| 14.8% | 50.0% | 57.5 | 1676 |
| 16.7% | 33.3% | **58.3** | 1698 |
| 22.2% | 0.0% | 52.7 | **1720** |

**(b) Global & local tokens.**

| Global | Local | GQA | MME |
|---|---|---|---|
| 0% | 100% | 58.0 | 1681 |
| 25% | 75% | 58.1 | 1689 |
| 50% | 50% | **58.3** | **1698** |
| 75% | 25% | 57.4 | 1688 |
| 100% | 0% | 57.5 | 1665 |

**(c) Local scan encoding layer $l$.**

| Encoding Layer | GQA | MME |
|---|---|---|
| $l = 2$ (*shallow*) | 57.1 | **1712** |
| $l = 6$ (*shallow*) | **58.3** | 1698 |
| $l = 12$ (*middle*) | 57.8 | 1692 |
| $l = 18$ (*deep*) | 57.5 | 1678 |
| $l = 23$ (*output*) | 57.3 | 1683 |

is able to preserve nearly the full performance of the original Video-LLaVA-7B. Moreover, it consistently surpasses prior methods across multiple benchmarks, highlighting the efficiency and robustness of our token reduction strategy in challenging video understanding tasks.

**Additional Results**. In Appendix A, we present further experimental results, demonstrating that our VScan generalizes to more challenging benchmarks that require fine-grained visual perception (*e.g.*, DocVQA [47] and InfoVQA [48]) and also effectively accelerates training while largely preserving model accuracy.

## 5.3 Ablation Studies

**Varying Retention Rates $R_1$ and $R_2$.** We analyze how different retention rate configurations affect performance by varying the retention rates $R_1$ and $R_2$ during the visual encoding and LLM decoding stages, respectively. As shown in Table 6 (a), we observe that relying solely on token selection in the visual encoder or applying overly aggressive pruning during LLM decoding leads to suboptimal performance. Specifically, text-agnostic pruning ($R_1$) reduces computation early but lacks awareness of the text prompt, making it prone to removing visually subtle yet instruction-critical cues. In contrast, text-aware pruning ($R_2$) benefits from intermediate cross-modal alignment and can better retain tokens relevant to the query, though pruning too aggressively at this stage may interrupt the model's semantic refinement and harm reasoning stability. Instead, a more balanced and gradual two-stage pruning strategy, *i.e.*, $R_1 = 16.7\%$ and $R_2 = 33.3\%$, achieves the best performance. These results also validate the effectiveness of combining both visual token reduction strategies to jointly improve efficiency and maintain accuracy.

**Mixing Global and Local Tokens**. In Table 6 (b), we examine the impact of mixing global and local token selections by varying their proportions. We find that selecting an equal ratio of global and local tokens yields the best performance, achieving 58.3% on GQA and 1698 on MME. In contrast, retaining only local or global tokens results in 0.3% and 0.8% performance drop on GQA, respectively. These results highlight the complementary roles of global and local tokens, which together capture rich visual information and help preserve the model's visual reasoning capabilities.

**Different Encoding Layers for Local Scan**. In Table 6 (c), we explore the effect of performing local token selection at different layers of the visual encoder. Consistent with our empirical findings in Section 3, we find that applying the local scan at a shallow layer ($l = 6$) yields the best performance. However, performing the local scan at very early ($l = 2$) or the output layer ($l = 23$) leads to a noticeable performance drop, with

performance on GQA falling to 57.1 and 57.3, respectively. This confirms that local tokens selected from shallow layers are more effective at capturing fine-grained visual details and better complement the global token set.

### 5.4 Efficiency Analysis

In Table 7, we evaluate the practical acceleration effects of VScan. Specifically, we report the total inference time and the time required for the pre-filling stage when processing all samples in the POPE benchmark. By retaining only 11% of the visual tokens, VScan achieves a 1.37× speedup in overall efficiency and a 1.77× speedup in prefilling efficiency on LLaVA-1.5-7B, while maintaining robust performance with only a 0.9% decline.

Our approach achieves even more significant acceleration on LLaVA-NeXT-7B, where it delivers a 2.05× speedup in inference and a 2.91× speedup in the prefill stage. Additionally, our approach can also effectively compress KV cache storage across different backbones. It is also important to note that VScan is compatible with FlashAttention [17], which can further enhance efficiency. For instance, the inference time of our approach with an 11% retention rate on LLaVA-NeXT-7B can be further reduced from 488 to 473 seconds.

Table 7: **Efficiency comparisons on the POPE benchmark**. We report the theoretical FLOPs, actual runtime, KV cache compression rate (%), and the achieved accuracy.

| Method | FLOPs ↓ | Total Time ↓ | Prefill Time ↓ | KV Cache ↓ | Accuracy ↑ |
|---|---|---|---|---|---|
| LLaVA-1.5-7B | 3.817 T | 1113 s (1.00×) | 416 s (1.00×) | 100.0% | 85.9 |
| + **Ours** (33%) | 1.253 T | 937 s (1.19×) | 301 s (1.38×) | 39.9% | **86.2** |
| + **Ours** (11%) | **0.415 T** | **812 s** (1.37×) | **235 s** (1.77×) | **19.9%** | 85.0 |
| LLaVA-NeXT-7B | 20.825 T | 2294 s (1.00×) | 1420 s (1.00×) | 100.0% | 86.5 |
| + **Ours** (33%) | 6.459 T | 1701 s (1.35×) | 994 s (1.43×) | 34.8% | **86.1** |
| + **Ours** (11%) | **2.099 T** | **1120 s** (2.05×) | **488 s** (2.91×) | **13.1%** | 85.1 |

## 6 Conclusion

In this work, we present a comprehensive empirical study to understand how visual information is processed across both the visual encoding and LLM decoding stages. Building on these insights, we propose VScan—a two-stage, training-free visual token reduction framework—to accelerate LVLM inference while maintaining robust performance. Specifically, we design complementary global and local scanning strategies to select informative visual tokens that preserve rich visual details during visual encoding, and further refine this token set via middle layer pruning in the LLM decoding stage based on textual relevance. Extensive experiments across 4 LVLM architectures and 16 image and video benchmarks demonstrate that our approach consistently outperforms existing state-of-the-art methods, achieving a superior trade-off between efficiency and accuracy.

**Limitations**. One potential limitation of this work is the inherent trade-off between efficiency and accuracy: while the proposed VScan significantly reduces inference cost of LVLMs, aggressive token pruning may still distort visual information and lead to degraded performance, particularly on challenging tasks that demand fine-grained understanding or compositional reasoning.

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

# VScan: Rethinking Visual Token Reduction for Efficient Large Vision-Language Models

## Appendix

In the appendix, we provide additional details and experimental results to enhance understanding and insights into our method. The appendix is organized as follows:

- Section A presents additional experimental results that further validate the effectiveness and robustness of our approach across various settings.

- Section B provides extended experimrntal details, including FLOPs calculation and full experimental configurations, to facilitate reproducibility.

- Section C lists the license information for all models, baselines, and benchmarks used in this work.

- Section D discusses the limitations of this work and explores its broader implications and impacts.

## A  Additional Experimental Results and Discussions

### A.1  Empirical Validation of Global and Local Scan

To validate the effectiveness of our global and local scan schemes, we construct adversarial subsets for GQA and POPE, namely AdvGQA and AdvPOPE, which contains failure cases similar to those shown in Figure 2 (*Left*), where the text queries play an important role and relying solely on the global scan to select visual tokens leads to errors. To accurately select these samples, we follow Gandelsman *et al.* [22] to decompose the image representations and pinpoint the tokens or regions most relevant to the query. Specifically, we utilize both the text and visual encoders of CLIP-ViT-L-336px [51] to identify the visual tokens relevant to the text query as a reference. Two examples of the visual tokens selected by CLIP are shown in Figure 2 (*Left*). A sample is included in the adversarial set if the response is correct when using the 64 tokens selected by CLIP, but becomes incorrect when using the 64 tokens selected solely by the global scan. Following this, we collected 886 and 515 adversarial samples in AdvGQA and AdvPOPE, respectively.

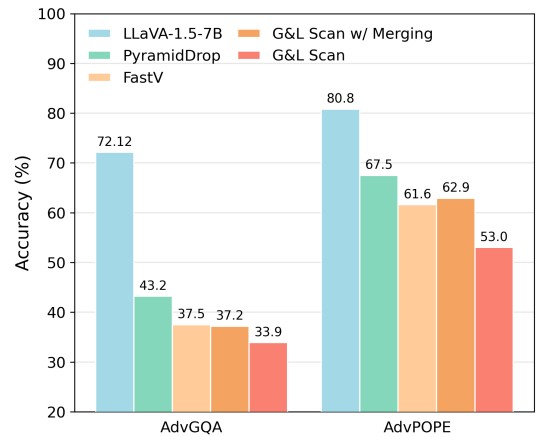

Figure A1: **Performance comparisons on AdvGQA and AdvPOPE**. We report the results for each approach with 64 visual tokens retained using LLaVA-1.5-7B [39].

In Figure A1, we present a performance comparison of incorporating both global and local scans with FastV [12] and PyramidDrop [62], which select visual tokens based on text attention and are expected to handle samples in the adversarial set effectively. We observe that incorporating both global and local scans achieves performance comparable to these text-guided approaches, despite being text-agnostic and selecting important tokens solely based on visual significance. These results validate that combining both scanning strategies and token merging helps preserve the maximum amount of visual information, effectively preventing information loss.

### A.2  Qualitative Results

In Figure A2, we present qualitative examples from the RefCOCO benchmark using Qwen-2.5-VL [5]. For each image, we show the model's predicted bounding boxes in response to different referring expressions, along with visualizations of the retained visual tokens after token pruning. These examples illustrate that our method can accurately localize the target objects described in queries, while significantly reducing the

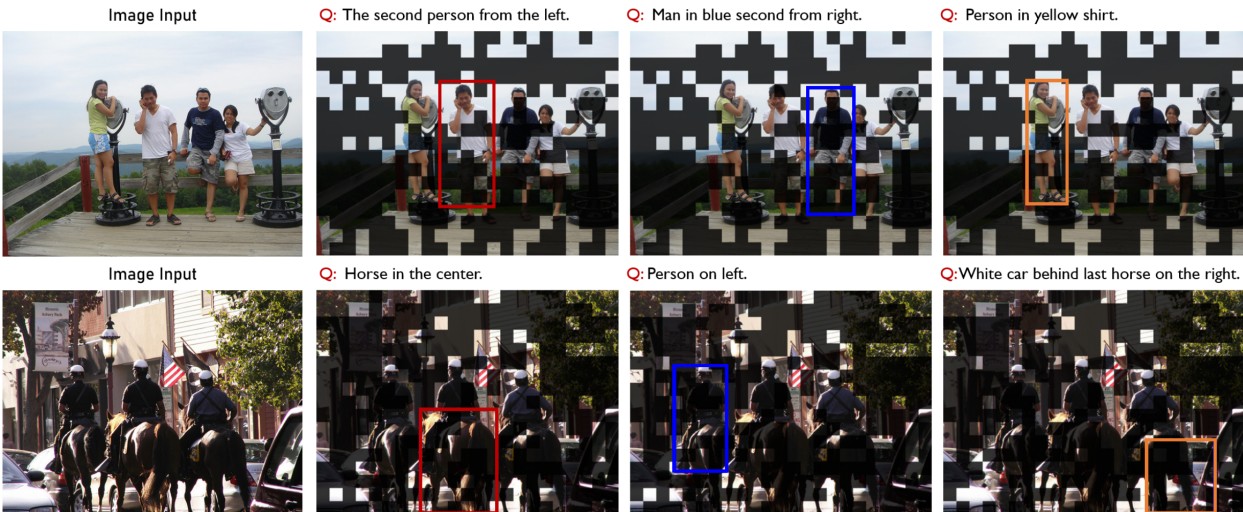

Figure A2: **Qualitative results on RefCOCO benchmark using Qwen-2.5-VL** [5]. We present the predicted boxes for 6 different queries on 2 images, along with visualizations of the retained tokens.

number of visual tokens used for inference. This demonstrates the model's ability to preserve semantically important information under token-efficient settings.

### A.3 Results on More Real-World Benchmarks

To comprehensively assess the effectiveness of VScan, we evaluate its performance on a range of real-world benchmarks, including DocVQA, InfoVQA, MME-RealWorld, MM-Vet, and MMMU, using the LLaVA-1.5-7B baseline. As shown in Table A1, VScan achieves competitive results while significantly reducing the number of visual tokens. With only 192 tokens, VScan maintains performance nearly identical to the full-token LLaVA-1.5-7B baseline, even surpassing it on MM-Vet and MMMU. When further reducing tokens to 128 and 64, VScan still preserves strong accuracy, demonstrating graceful degradation across tasks. These results highlight the robustness and efficiency of our method, showing that substantial token reduction can be achieved without sacrificing performance on diverse real-world benchmarks.

We further compare VScan with FastV and PDrop under different token retention rates on DocVQA and InfoVQA, using both LLaVA-NeXT-7B and Qwen2.5-VL-7B backbones. As shown in Table A2, VScan consistently outperforms FastV and PDrop at the same retention levels. For LLaVA-NeXT-7B, VScan achieves 72.6 on DocVQA and 36.3 on InfoVQA with only 33% of tokens, closely matching the full-model baseline (74.4/37.1) while retaining higher accuracy than FastV and PDrop. Similar trends hold at more aggressive pruning levels, where VScan shows lower degradation compared to stronger drops for the baselines. Results on Qwen2.5-VL-7B further confirm this advantage: VScan reaches 94.0/79.7 at 33% retention and maintains 83.9/73.4 even at 11%, substantially outperforming FastV and PDrop. These results demonstrate that VScan generalizes across architectures and provides a more robust trade-off between efficiency and accuracy.

Table A1: **Performance of our VScan across diverse real-world benchmarks**. We report results on DocVQA, InfoVQA, MME-RealWorld, MM-Vet, and MMMU. Compared with the LLaVA-1.5-7B baseline, VScan achieves competitive performance while substantially reducing the number of visual tokens retained.

| Method | Retention Rate | DocVQA | InfoVQA | MME-RealWorld | MM-Vet | MMMU |
|--------|---------------|--------|---------|---------------|--------|------|
| LLaVA-1.5-7B | – | 28.1 | 25.8 | 24.9 | 31.1 | 35.3 |
| VScan | 33.3% | 27.5 | 25.7 | 24.0 | 31.8 | 35.7 |
| VScan | 22.2% | 25.7 | 25.4 | 22.7 | 30.5 | 36.1 |
| VScan | 11.1% | 23.9 | 23.7 | 22.2 | 29.7 | 35.8 |

Table A2: **Comparison of VScan with FastV and PDrop on DocVQA and InfoVQA.** Results are reported under different token retention rates (33%, 22%, 11%). †Evaluation is based on our re-implementation.

| Method | DocVQA | InfoVQA |
|---|---|---|
| LLaVA-NeXT-7B | 74.4 | 37.1 |
| FastV† (33%) | 69.8 | 33.5 |
| PDrop† (33%) | 71.8 | 35.0 |
| VScan (33%) | 72.6 | 36.3 |
| FastV† (22%) | 64.7 | 32.8 |
| PDrop† (22%) | 67.4 | 33.4 |
| VScan (22%) | 71.6 | 34.2 |
| FastV† (11%) | 61.2 | 28.1 |
| PDrop† (11%) | 64.0 | 31.6 |
| VScan (11%) | 68.8 | 34.5 |

(a) LLaVA-NeXT-7B.

| Method | DocVQA | InfoVQA |
|---|---|---|
| Qwen2.5-VL-7B | 95.7 | 82.6 |
| FastV† (33%) | 88.7 | 75.3 |
| PDrop† (33%) | 90.1 | 76.2 |
| VScan (33%) | 94.0 | 79.7 |
| FastV† (22%) | 81.6 | 70.8 |
| PDrop† (22%) | 83.8 | 73.7 |
| VScan (22%) | 87.8 | 77.1 |
| FastV† (11%) | 74.9 | 65.2 |
| PDrop† (11%) | 77.4 | 67.0 |
| VScan (11%) | 83.9 | 73.4 |

(b) Qwen2.5-VL-7B.

## A.4  VScan for Accelerating Training

We further validate the effectiveness of VScan in reducing training cost while preserving the model performance. As shown in Table A3, VScan reduces training time by 41.7% from 96 GPU hours to 56 GPU hours, achieving 96.7% of the original performance. For comparison, we also report inference-time VScan results, which reach 95.2% of baseline performance. Notably, applying VScan during training yields a 1.5% improvement over using VScan solely at inference.

Table A3: **Training efficiency and performance of VScan compared with the LLaVA-1.5-7B baseline**. VScan significantly reduces GPU hours while maintaining strong performance across benchmarks.

| Method | GPU Hours | GQA | MMB | MMB-CN | MME | POPE | Avg. |
|---|---|---|---|---|---|---|---|
| LLaVA-1.5-7B | 96 | 61.9 | 64.7 | 58.1 | 1862 | 85.9 | 100.0% |
| VScan (fine-tune) | 56 | 59.2 | 63.0 | 57.9 | 1721 | 84.6 | 96.7% |
| VScan (inference) | – | 58.3 | 62.1 | 55.7 | 1698 | 85.0 | 95.2% |

## A.5  Remarks on Multi-Turn Conversations

Adapting the middle-layer pruning component of VScan to support multi-turn conversations is straightforward: When presented with new questions, VScan can reassess token importance and reselect textually relevant visual tokens from the existing token pool through global-local scans. Although this re-selection introduces additional computation compared to text-agnostic methods (which are inherently compatible with multi-turn dialogue), the overhead is negligible relative to the overall prefill time.

## A.6  More Efficiency Comparisons

Table A4 summarizes the trade-off between efficiency and performance for various visual-token pruning methods across different retention ratios (33.3%, 22.2%, and 11.1%). VScan consistently achieves superior accuracy while maintaining competitive FLOPs and inference time, demonstrating its effectiveness in preserving visual information under aggressive pruning.

## A.7  Full Numerical Results on Qwen-2.5-VL-7B

We report the full numerical results on Qwen-2.5-VL-7B in Tables A5, A6, and A7.

Table A4: **Efficiency and performance comparison under different token retention ratios.** We report the total FLOPs (T), inference time (s), and accuracy achieved by the model at a given retention rate.

| Method | 33.3% | 22.2% | 11.1% |
|---|---|---|---|
| LLaVA-1.5-7B | - | 3.817T / 416s / 85.9 | - |
| FastV | 1.253T / 298s / 60.7 | 0.836T / 266s / 57.2 | 0.421T / 231s / 44.5 |
| PDrop | 1.253T / 307s / 82.3 | 0.834T / 278s / 82.3 | 0.415T / 240s / 55.9 |
| VisionZip | 1.253T / 293s / 85.3 | 0.834T / 263s / 83.2 | 0.415T / 229s / 77.0 |
| VScan | 1.253T / 301s / 86.2 | 0.834T / 274s / 86.1 | 0.415T / 235s / 85.0 |

Table A5: **MMBench accuracy** under different visual token retention ratios. Higher is better.

| Model | Method | 11% | 22% | 33% | 50% | 100% |
|---|---|---|---|---|---|---|
| | FastV | 70.13 | 78.09 | 81.01 | 85.40 | 87.37 |
| **32B** | PDrop | 73.22 | 79.04 | 82.74 | 86.03 | 87.37 |
| | Ours | **78.09** | **82.47** | **85.31** | **86.17** | **87.37** |
| | FastV | 69.07 | 74.74 | 77.73 | 80.27 | 84.19 |
| **7B** | PDrop | 71.08 | 75.43 | 78.04 | 81.96 | 84.19 |
| | Ours | **76.89** | **80.50** | **80.67** | **82.56** | **84.19** |
| | FastV | 56.79 | 67.27 | 72.42 | 74.98 | 78.87 |
| **3B** | PDrop | 55.93 | 63.22 | 70.30 | 72.48 | 78.87 |
| | Ours | **69.67** | **70.62** | **72.81** | **75.64** | **78.87** |

# B More Experimental Details

## B.1 Remarks on Ensuring Fair Comparisons

In our main result table, we ensured fair comparisons between our VScan and text-agnostic approaches such as VisionZip. Instead of aligning on the final token retention ratio, we matched the average token retention across all LLM layers, which roughly correlates with the inference complexity/speed. For example, at 11.1% retention rate, VScan retains 96 tokens before the LLM using a global-local scan strategy and then prunes to 32 tokens at the middle LLM layer (layer 16 of 32). This yields an average of 64 tokens retained across all LLM layers—matching the average retention of the text-agnostic methods, ensuring that inference speed is consistent.

Additionally, in the comparisons, the compared text-aware approaches, such as FastV and SparseVLM, actually have an advantage over our method, as their reported results are based on the final token count. Since we were unable to fully reproduce all of these methods, we directly reported their results as presented in their respective papers.

To account for this, in the Qwen-2.5-VL and Video-LLaVA experiments presented, we reproduce FastV and PyramidDrop while aligning the average token retention across all LLM layers to ensure a fair comparison (marked in the tables with a †). For instance, for FastV, to achieve an average retention rate of 11%, we prune to 5.2% of the original token count starting from LLM layer 2, resulting in an average retention of $(100\% \times 2 + 5.2\% \times 30) / 32 = 11.1\%$.

Therefore, under the same average retention rate, the efficiency metrics are essentially similar. This means that, at the same retention rate, higher performance directly translates into a more favorable performance–efficiency trade-off. In this regard, we have conducted thorough experiments demonstrating that our approach achieves a state-of-the-art performance–efficiency trade-off across benchmarks and models.

Table A6: **MMBench-CN accuracy** under different visual token retention ratios. Higher is better.

| Model | Method | 11% | 22% | 33% | 50% | 100% |
|-------|--------|-----|-----|-----|-----|------|
| **32B** | FastV | 67.61 | 76.12 | 79.81 | 83.42 | 86.00 |
| | PDrop | 71.31 | 77.65 | 80.72 | 84.28 | 86.00 |
| | Ours | **77.23** | **80.76** | **84.45** | **84.78** | **86.00** |
| **7B** | FastV | 66.92 | 70.53 | 76.98 | 77.41 | 81.79 |
| | PDrop | 68.59 | 69.85 | 76.12 | 78.15 | 81.79 |
| | Ours | **74.48** | **76.37** | **78.69** | **79.04** | **81.79** |
| **3B** | FastV | 54.73 | 66.32 | 71.13 | 74.05 | 77.41 |
| | PDrop | 57.20 | 65.92 | 71.58 | 74.79 | 77.41 |
| | Ours | **63.66** | **68.47** | **72.45** | **74.51** | **77.41** |

Table A7: **MME total scores** under different visual token retention ratios. Higher is better.

| Model | Method | 11% | 22% | 33% | 50% | 100% |
|-------|--------|-----|-----|-----|-----|------|
| **32B** | FastV | 1536 | 1924 | 2124 | 2316 | 2432 |
| | PDrop | 1536 | 1797 | 2078 | 2279 | 2432 |
| | Ours | **2015** | **2189** | **2288** | **2378** | **2432** |
| **7B** | FastV | 1970 | 1990 | 2240 | 2257 | 2291 |
| | PDrop | 1982 | 2064 | 2218 | 2236 | 2291 |
| | Ours | **1990** | **2135** | **2275** | **2270** | **2291** |
| **3B** | FastV | 1620 | 1800 | 1953 | 2053 | 2139 |
| | PDrop | 1444 | 1692 | 1709 | 1986 | 2139 |
| | Ours | **1763** | **1838** | **1977** | **2037** | **2139** |

## B.2 Computational Complexity

We follow PyramidDrop [62] to compute the theoretical floating-point operations (FLOPs) introduced in the LLM decoding layers during the pre-filling stage for processing the visual tokens. Specifically, in each of the $K$ decoding layers, self-attention calculation with a causal mask is applied, followed by multiple feed-forward network (FFN) layers. The total FLOPs can thus be computed as:

$$\text{Total FLOPs} = \sum_{k=1}^{K} \left( 4n_k d^2 + 2n_k^2 d + 3n_k dm \right), \tag{6}$$

where $K$ is the number of transformer layers, $n_k$ is the number of visual tokens at LLM layer $k$, $d$ is the hidden state size, and $m$ is the intermediate size of the FFN. This calculation suggests that reducing the number of visual tokens can significantly decrease the FLOPs required during inference.

## B.3 Experimental Design for Study 3

The goal of Study 3 is to determine at which LLM layers the next-token predictions stabilize, i.e., when the model's forward computation becomes semantically saturated. Once the prediction has converged, further layers mainly refine numerical precision rather than introduce new semantic information. Understanding this stabilization point is crucial because pruning before convergence can disrupt semantic formation, whereas pruning after convergence is largely safe and provides meaningful computational speedups.

Here is our experimental setup: For each decoding layer $l$ in the LLaVA-1.5-7B LLM backbone, we extract its hidden state $h_l$ and pass it through the original final linear projection layer to obtain the logits, i.e., $s_l = W_{\text{proj}} h_l$. We then apply a softmax to compute the prediction confidence and report the top-confidence token for each layer.

---

You are an intelligent chatbot designed for evaluating the correctness of generative outputs for question-answer pairs. Your task is to compare the predicted answer with the correct answer and determine if they match meaningfully. Here's how you can accomplish the task:

##INSTRUCTIONS:
- Focus on the meaningful match between the predicted answer and the correct answer.
- Consider synonyms or paraphrases as valid matches.
- Evaluate the correctness of the prediction compared to the answer.

---

**User Input:**
Please evaluate the following video-based question-answer pair:

Question: {question}
Correct Answer: {answer}
Predicted Answer: {pred}

Provide your evaluation only as a yes/no and score where the score is an integer value between 0 and 5, with 5 indicating the highest meaningful match. Please generate the response in the form of a Python dictionary string with keys 'pred' and 'score', where value of 'pred' is a string of 'yes' or 'no' and value of 'score' is in INTEGER, not STRING. DO NOT PROVIDE ANY OTHER OUTPUT TEXT OR EXPLANATION. Only provide the Python dictionary string. For example, your response should look like this: {'pred': 'yes', 'score': 4}.

---

Table B8: **GPT-aided evaluation setup**. We present the prompt and user input format for evaluating the LVLM responses in video understanding tasks.

### B.4 GPT-Aided Evaluation on Video Understanding Tasks

Following the evaluation protocol of Video-LLaVA, we employ GPT-3.5-Turbo to assess model responses on video understanding tasks, evaluating them based on both accuracy and quality score. Specifically, we adopt the prompt shown in Table B8 to guide GPT in rating each response:

- **Accuracy**: A binary yes/no judgment indicating whether the response is correct.

- **Score**: An integer ranging from 0 to 5, where 5 represents the highest degree of relevance and informativeness and indicates the highest meaningful match.

## C License Information

We list the license information for all the used assets as follows.

**Benchmarks.** We evaluate on a comprehensive set of 16 benchmarks spanning image QA, video QA, multimodal reasoning, and referential understanding tasks:

- **GQA** [25]: compositional visual question answering. This benchmark is released under the CC BY 4.0 license.

- **ScienceQA** [43]: multimodal science questions with diagrams and text. This benchmark is released under the MIT license.

- **VQAv2** [23]: real-world image-based QA with balanced answers. This benchmark is released under the CC BY 4.0 license.

- **TextVQA** [53]: reading and reasoning over scene text in images. This benchmark is released under the CC BY 4.0 license.

- **VizWiz** [24]: real-world visual questions from blind users. This benchmark is released under the CC BY 4.0 license.
- **MMBench** [42] / **MMBench-CN** [42]: multilingual multimodal reasoning. These benchmarks are released under Apache License 2.0.
- **MME** [20]: fine-grained multimodal evaluation on object, OCR, and commonsense. This benchmark is released under Apache License 2.0.
- **POPE** [35]: probing object hallucinations in vision-language models. This benchmark is released under the MIT license.
- **RefCOCO** / **RefCOCO+** [30], **RefCOCOg** [46]: referential expression grounding. These benchmarks are released under Apache License 2.0.
- **TGIF** [27]: video QA with spatiotemporal reasoning. The license of this work is not specified.
- **MSVD** [10]: short video captioning and QA. This benchmark is released under the MIT license.
- **MSRVTT** [63]: large-scale video-text retrieval and QA. This benchmark is released under the MIT license.
- **ActivityNet-QA** [66]: complex event-centric video QA. This benchmark is released under Apache License 2.0.

**Models**. We apply our approach to four widely used LVLMs.

- **LLaVA-1.5** [39] is released under the LLaMA 2 Community License.
- **LLaVA-NeXT** [40] is released under Apache License 2.0.
- **Qwen-2.5-VL** [5] is released under Apache License 2.0.
- **VideoLLaVA** [36] is released under Apache License 2.0.

**Code**. Our codebase builds upon PyramidDrop [62], licensed under MIT, and FasterVLM [70], licensed under Apache 2.0.

# D  Limitations and Broader Impacts

**Limitations**. One key limitation of this work is the inherent trade-off between efficiency and accuracy: while the proposed VScan significantly reduces inference cost of LVLMs, aggressive token pruning may still distort visual information and lead to degraded performance, particularly on challenging tasks that demand fine-grained understanding or compositional reasoning.

**Broader Impacts**. The development of efficient LVLMs has significant potential to influence a wide range of applications, from autonomous systems and robotics to healthcare, education, and accessibility. By optimizing visual token reduction with VScan, we are addressing the computational overheads associated with processing large visual inputs, enabling faster and more efficient inference in real-time applications. This can lead to more widespread adoption of LVLMs in settings where rapid decision-making is crucial, such as autonomous vehicles, real-time video analysis, and interactive AI systems.

