# OpenReview forum: "VScan: Rethinking Visual Token Reduction for Efficient Large Vision-Language Models"
_TMLR — Accepted by TMLR_

### Review · Reviewer_AvVV · 2025-10-28

**Summary Of Contributions:**

In this paper, the authors analyze existing efficiency VLM methods and propose VScan, a two-stage visual token reduction framework. The first stage integrates complementary global and local scans with token merging during visual encoding; the second stage prunes tokens at intermediate layers of the language model. Experiments across four VLMs show that VScan accelerates inference and outperforms strong baselines, while maintaining competitive accuracy.

**Audience:**

Yes

**Audience Explanation:**

The topic of Efficient VLMs is highly relevant to real-world applications and has gained significant attention in recent years.

**Broader Impact Concerns:**

None concern

**Claims And Evidence:**

Yes

**Claims Explanation:**

As a researcher in this field, I have observed similar findings to those reported by the authors; therefore, I find the paper both convincing and clearly presented.

**Requested Changes:**

1. Firstly, I think the authors’ empirical analysis is very interesting and well written, and I believe these analyses could further advance research in this field. However, regarding **Study 3: At which LLM layer do next-token predictions begin to converge?**, I am unclear about its purpose, experimental design, how the conclusions were derived, and how these conclusions contribute to subsequent method design. I hope the authors can elaborate further.

2. I fully understand that due to the use of the Patch Merger, the improvement of the Efficient VLM method on the Qwen2.5VL model might not be very significant. Nevertheless, I still encourage the authors to present the Qwen2.5VL results in a **table format similar to Table 4**, rather than only using a line chart, as I believe this would better promote progress and development in the field.

3. Since the entire method consists of the **text-agnostic** part in Section 4.1 and the **text-aware** part proposed in Section 4.2, I hope the ablation study can further clarify how each part respectively affects **speed** and **performance**. A deeper discussion of their respective advantages and disadvantages would make the paper much more meaningful.

4. For **Table 5, Table A2, and Table A3**, I think that since the authors discussed many **text-agnostic** and **text-aware** methods in the introduction and writing of the paper, I suggest that in a future version, the benchmarks should also include **both categories** of methods, rather than comparing only one. This would more convincingly demonstrate the effectiveness of the proposed approach.

5. In paper [1, 2], the researchers discussed that in most general scenarios, even simple resizing can achieve strong performance. How do the authors view this issue? I look forward to some discussion on this point.

[1] VisionThink: Smart and Efficient Vision Language Model via Reinforcement Learning

[2] Are We Using the Right Benchmark: An Evaluation Framework for Visual Token Compression Methods

---

> ### Author Response · Authors · 2025-12-03
> **Response to Reviewer AvVV (Part 1/2)**
>
> Dear Reviewer AvVV,
>
> Thank you for your insightful comments on our work. We provide point-by-point responses to address your concerns below.
>
> ---
>
> **Comment (1)**: “*Firstly, I think the authors’ empirical analysis is very interesting… However, regarding Study 3: At which LLM layer do next-token predictions begin to converge? I am unclear about its purpose, experimental design, how the conclusions were derived, and how these conclusions contribute to subsequent method design. I hope the authors can elaborate further*.”
>
> **Response (1)**: Thank you very much for your constructive comments and for the positive assessment of our empirical analyses. Below we clarify the purpose, experimental design, how conclusions were derived, and most importantly, how these findings inform the design of our middle-layer pruning strategy.
>
> - **Purpose**: The goal of Study 3 is to determine at which LLM layers the next-token predictions stabilize, i.e., when the model’s forward computation becomes semantically saturated. Once the prediction has converged, further layers mainly refine numerical precision rather than introduce new semantic information. Understanding this stabilization point is crucial because pruning before convergence can disrupt semantic formation, whereas pruning after convergence is largely safe and provides meaningful computational speedups.
>
> - **Experimental Design**: For each decoding layer $l$ in the LLaVA-1.5-7B LLM backbone, we extract its hidden state $h_l$ and pass it through the original final linear projection layer to obtain the logits, i.e., $s_l = W_{\text{proj}} h_l$. We then apply a softmax to compute the prediction confidence and report the top-confidence token for each layer.
>
> - **Conclusions**: We observe that in more challenging open-ended tasks like GQA, the next-token predictions stabilize around LLM layer 20, whereas in simpler yes/no tasks such as POPE, the predictions converge earlier, around LLM layer 16.
>
> - **How This Informs Our Approach**: Our findings indicate that early layers are still forming core cross-modal semantics, and pruning them risks disrupting essential grounding. In contrast, by the middle layers, next-token predictions have largely stabilized, meaning that these layers contribute diminishing semantic change. This directly motivates pruning in the middle-to-late layers rather than the early layers.
>
> We have expanded Study 3 in the revised paper to include the reasoning, and we have also added a more detailed description of the experimental design in Appendix B.3.
>
> ---
>
> **Comment (2)**: “*I still encourage the authors to present the Qwen2.5VL results in a table format similar to Table 4, rather than only using a line chart, as I believe this would better promote progress and development in the field.*”
>
> **Response (2)**:  We agree with your point and following your comments, we have included the full numerical results on Qwen-2.5-VL in table format in Appendix A.7 of the revised paper.
>
> ---
>
> **Comment (3)**: “*Since the entire method consists of the text-agnostic part in Section 4.1 and the text-aware part proposed in Section 4.2, I hope the ablation study can further clarify how each part respectively affects speed and performance. A deeper discussion of their respective advantages and disadvantages would make the paper much more meaningful.*”
>
> **Response (3)**: We would like to refer the reviewer to the ablation study in Table 6(a), where we vary the pruning ratios at the two stages while keeping the overall pruning ratio fixed at 11.1\% to ensure comparable speed. When we set $R_2 = 100\%$, the method degrades to pure text-agnostic pruning, which leads to suboptimal performance. Conversely, reducing $R_2$ too aggressively results in overly strong text-aware pruning that also harms accuracy. A more balanced and gradual two-stage strategy, specifically $R_1$ = 16.7% and $R_2$ = 33.3%, achieves the best results and highlights the importance of combining both stages.
>
> To offer deeper insight, we further analyze the respective strengths and limitations of the two components. Text-agnostic pruning ($R_1$) reduces computation early but lacks awareness of the text prompt, making it prone to removing visually subtle yet instruction-critical cues. In contrast, text-aware pruning ($R_2$) benefits from intermediate cross-modal alignment and can better retain tokens relevant to the query, though pruning too aggressively at this stage may interrupt the model’s semantic refinement and harm reasoning stability. Following your suggestion, we have now incorporated a more detailed discussion of these advantages and disadvantages in Section 5.3 of the revised paper.

---

> ### Author Response · Authors · 2025-12-03
> **Response to Reviewer AvVV (Part 2/2)**
>
> **Comment (4)**: “*For Table 5, Table A2, and Table A3, I think that since the authors discussed many text-agnostic and text-aware methods in the introduction and writing of the paper, I suggest that in a future version, the benchmarks should also include both categories of methods, rather than comparing only one. This would more convincingly demonstrate the effectiveness of the proposed approach.*”
>
> **Response (4)**: Following your suggestion, we have additionally re-implemented VisionZip, a representative text-agnostic pruning method, and incorporated its results into Tables 4 and 5 of the main paper. This provides a more complete comparison between text-agnostic and text-aware approaches and further supports the effectiveness of our method.
>
> ---
>
> **Comment (5)**: “*In paper [1, 2], the researchers discussed that in most general scenarios, even simple resizing can achieve strong performance. How do the authors view this issue? I look forward to some discussion on this point.*”
>
> **Response (5)**: Thanks for bringing these excellent works to our attention. In our experiments, we observed results consistent with [1]: on general benchmarks, simple resizing baselines (e.g., reducing resolution to one quarter) can outperform FastV, but they typically perform slightly worse than PDrop and also worse than our approach (at a 22.2% pruning ratio). On more challenging benchmarks, such as those presented in Table A1 where fine-grained visual details are crucial for correct reasoning, the resizing baseline underperforms our method by a substantial margin. This shows that while resizing can handle easier cases, it struggles to preserve the fine-grained information needed for more complex tasks. (PS: Our experimental results do not fully align with [2]; we find that downsampling can outperform FastV but still underperforms other approaches, such as VisionZip.)
>
> The authors consider simple resizing, along with other vanilla strategies such as random token selection, as baseline techniques that any proposed method must outperform in order to demonstrate meaningful utility in real-world settings. While effective and easy to implement, simple resizing has two clear limitations: (1) Compared to text-agnostic approaches, it reduces visual information uniformly and may discard fine-grained details that dedicated token-selection methods (such as VisionZip) tend to preserve. (2) Compared to text-aware approaches, simple resizing completely ignores the text instruction and therefore cannot adaptively retain regions that are semantically relevant to the query.
>
> Compared to these baselines, our VScan introduces a novel two-stage, training-free visual token reduction framework that leverages the strengths of two lines of approaches, and enhances the efficiency of LVLMs by progressively pruning uninformative tokens during both visual encoding and language decoding stages. We also hope that our empirical findings help highlight certain limitations of existing paradigms and offer perspectives that could be useful for future research in this field.
>
> ---
>
> We hope that our responses have addressed your concerns. If you have additional comments or concerns, please let us know and we will be more than happy to answer.
>
> Best,
>
> Authors

---

> > ### Comment · Reviewer_AvVV · 2025-12-28
> > **Thanks for the rebuttal**
> >
> > Thanks for the rebuttal, the responses addressed my concerns, I will give a positive score. Besides, the analysis in Response (5) is valuable; I recommend incorporating it into the main paper to offer insights for future research.

---

> > > ### Author Response · Authors · 2026-01-04
> > > **Thank you for your constructive review**
> > >
> > > We are glad that our rebuttal helped address your earlier concerns. Following your suggestion, we will incorporate Response (5) into the main paper in the next revision. Your feedback has been very helpful in strengthening our work, and we sincerely appreciate your time and constructive comments.

---

### Review · Reviewer_YT24 · 2025-11-11

**Summary Of Contributions:**

This paper presents VScan, a general and dynamic approach for compressing visual tokens in Large Vision-Language Models (LVLMs). Unlike most prior works on visual token compression, VScan is grounded in thorough empirical analysis and addresses two key limitations of existing methods:

Over-reliance on global compression: Previous methods typically perform a global token reduction based solely on the CLS token’s attention weights from the vision transformer. Such global reliance often overlooks fine-grained visual details, which are crucial for tasks like grounding-oriented visual question answering (VQA).

Lack of text-aware guidance: Many existing approaches conduct compression purely within the vision modality, neglecting the important role of textual context in determining which visual tokens are semantically relevant.

To overcome these issues, VScan introduces a two-stage compression framework. In the first stage, it performs an iterative global–local scan within the vision transformer, pruning redundant visual tokens while retaining both key global representations and essential local details. In the second stage, it applies text-guided filtering by leveraging attention between the remaining visual tokens and the final text guidance tokens, further removing redundant visual information.

Extensive experiments and ablation studies demonstrate that VScan consistently outperforms recent visual compression methods for LVLMs, maintaining comparable or even superior performance to non-compressed baselines at equivalent compression ratios.

**Audience:**

Yes

**Audience Explanation:**

As discussed above, VScan is a well-motivated and generally applicable approach to visual token compression in LVLMs. It demonstrates strong performance across standard benchmarks, and with the authors’ commitment to releasing the code upon acceptance, VScan has the potential to serve as a solid baseline for future research in this area.

**Broader Impact Concerns:**

No concerns are identified regarding the ethical implications of this work.

**Claims And Evidence:**

Yes

**Claims Explanation:**

The methodology of VScan is well-motivated by comprehensive empirical analyses. The authors first demonstrate that as layers in the vision transformer deepen, both the [CLS] attention maps and the self-attention maps of representative visual tokens exhibit a local-to-global transition. This observation reveals that relying solely on the [CLS] token from the final layer causes the model to lose sensitivity to local visual details. Furthermore, the text-guided filtering stage is well-justified through analyses of LLM position bias and attention patterns, which motivate the design choice of using mid-level LLM layers for refining and pruning vision tokens. Finally, the effectiveness of VScan is thoroughly validated through extensive experiments and ablation studies on widely used benchmarks, consistently demonstrating its advantages over existing baselines in visual token compression for LVLMs.

**Requested Changes:**

Questions for Clarification and Discussion

1. Compatibility with KV Cache and Inference Optimization:
Is VScan compatible with commonly used inference optimizations such as KV cache and FlashAttention? My understanding is that, although the proposed method effectively reduces the number of visual tokens, it may alter the cache structure used during autoregressive decoding. This could potentially limit the benefits of token reduction, as it might break cache reuse and lead to higher inference overhead or memory consumption. It would be helpful if the authors could clarify whether VScan can be integrated with these optimizations in practice, and if not, to discuss the trade-offs and possible solutions.

2. Choice and Scope of Evaluation Benchmarks:
While evaluating VScan on standard benchmarks is appropriate for comparison with prior works, the field of vision-language modeling has rapidly evolved, and many existing benchmarks have become less discriminative due to the increasingly strong text priors of modern LLMs. In some cases, these benchmarks can be solved largely from text alone, diminishing the significance of visual compression performance. Therefore, I would encourage the authors to discuss or explore more challenging, spatially grounded benchmarks that heavily depend on visual understanding. This would better demonstrate the extent to which VScan preserves true vision-dependent performance and highlight its strengths beyond simple compression gains.

---

> ### Author Response · Authors · 2025-12-03
> **Response to Reviewer YT24**
>
> Dear Reviewer YT24,
>
> Thanks for your valuable feedback! We provide point-by-point responses to address your concerns below.
>
> ---
>
> **Comment (1)**: “*Compatibility with KV Cache and Inference Optimization: Is VScan compatible with commonly used inference optimizations such as KV cache and FlashAttention? My understanding is that, although the proposed method effectively reduces the number of visual tokens, it may alter the cache structure used during autoregressive decoding.*”
>
> **Response (1)**: We would like to clarify that our VScan is compatible with both KV cache and FlashAttention. Here are the detailed clarifications:
>
> - We acknowledge that VScan’s token-pruning process affects the KV cache. However, pruning occurs **before** the visual tokens are added to the cache. As a result, the KV cache simply stores fewer entries, while its structure and format remain unchanged. Thus, VScan is fully compatible with standard KV caching mechanisms. Moreover, as shown in Table 7, VScan can even reduce KV cache storage across different backbones, further demonstrating its practical effectiveness.
>
> - Yes, VScan is compatible with FlashAttention, as discussed in Section 4.2. In our implementation, we recompute the attention scores for the final instruction token using a standard (non-Flash) attention operation outside the standard LLM layers. We will release the code upon acceptance to fully illustrate our solution.
>
> ---
>
> **Comment (2)**: “*I would encourage the authors to discuss or explore more challenging, spatially grounded benchmarks that heavily depend on visual understanding. This would better demonstrate the extent to which VScan preserves true vision-dependent performance and highlight its strengths beyond simple compression gains.*.”
>
> **Response (2)**: Thanks for pointing out this. Yes, we agree with your point. We would like to refer you to the experimental results in Appendix A.3, where we extensively evaluate our approach on five highly visual-centric tasks, including DocVQA and InfoVQA for OCR, as well as MME-RealWorld, which contains challenging scenarios such as OCR in the wild and remote sensing. We also attached the results here:
>
> |Method (#Tokens)|DocVQA|InfoVQA|MME-RealWorld|MM-Vet|MMMU|
> |-|:-:|:-:|:-:|:-:|:-:|
> |LLaVA-1.5-7B (576)|28.1|25.8|24.9|31.1|35.3|
> |VScan (192)|27.5|25.7|24.0|31.8|35.7|
> |VScan (128)|25.7|25.4|22.7|30.5|36.1|
> |VScan (64)|23.9|23.7|22.2|29.7|35.8|
>
> While our approach shows a slight performance drop on these more challenging fine-grained tasks compared to general VQA, it still demonstrates strong performance. Even at an aggressive retention rate of 11.1%, VScan maintains over 85% of the original performance. With a moderate retention rate (e.g., 33%), it achieves up to 95%, highlighting the visual token redundancy even in fine-grained settings and validating the effectiveness of our method.
>
> We further compare our approach with FastV and PDrop on challenging DocVQA and InfoVQA benchmarks. The results are as follows:
>
> | Method | DocVQA | InfoVQA |
> |-|:-:|:-:|
> | LLaVA-NeXT-7B| 74.4   | 37.1  |
> | FastV (33%)| 69.8   | 33.5    |
> | PDrop (33%)| 71.8   | 35.0    |
> | VScan (33%)| **72.6**   | **36.3**    |
> | FastV (22%)| 64.7   | 32.8    |
> | PDrop (22%)| 67.4  | 33.4    |
> | VScan (22%)| **71.6**   | **34.2**    |
> | FastV (11%) | 61.2   | 28.1    |
> | PDrop (11%) | 64.0   | 31.6    |
> | VScan (11%) |**68.8**   | **34.5**    |
>
> | Method| DocVQA | InfoVQA |
> |-|:-:|:-:|
> | Qwen2.5-VL-7B | 95.7| 82.6 |
> | FastV (33%)| 88.7   | 75.3   |
> | PDrop (33%)| 90.1   | 76.2    |
> | VScan (33%) | **94.0** | **79.7** |
> | FastV (22%)| 81.6   | 70.8    |
> | PDrop (22%)| 83.8   | 73.7   |
> | VScan (22%) | **87.8** | **77.1** |
> | FastV (11%)| 74.9   | 65.2    |
> | PDrop (11%)| 77.4   | 67.0    |
> | VScan (11%) | **83.9** | **73.4** |
>
> We can see that our VScan consistently outperforms both PDrop and FastV across various settings on these fine-grained benchmarks.
>
> Additionally, we would like to highlight that our paper includes experimental results on highly challenging referring-grounding tasks that require precise spatial understanding, evaluated using the advanced Qwen-2.5-VL-7B model. These results further demonstrate the effectiveness of our approach beyond general QA tasks.
>
> ---
>
>
> If you have additional comments or concerns, please let us know and we will be more than happy to answer.
>
> Best,
>
> Authors

---

> > ### Comment · Reviewer_YT24 · 2025-12-30
> > **Thanks for the rebuttal**
> >
> > Thanks authors for the detailed rebuttal and new experimental results which resolved my concerns. I don't have any further questions and will recommend acceptance.

---

> > > ### Author Response · Authors · 2026-01-04
> > > **Thank you for your constructive review**
> > >
> > > Thank you very much for taking the time to read our rebuttal. We truly appreciate your careful consideration and are glad our response addressed your concerns.

---

### Review · Reviewer_k8qv · 2025-11-22

**Summary Of Contributions:**

**Summary**

This paper addresses the high computational cost of recent LVLMs, which arises from increasingly long visual token sequences produced by fine-grained visual encoding. The authors first conduct an empirical analysis of how visual tokens are processed across both the vision encoder and the language model, revealing limitations in existing pruning strategies that operate either only at the encoder output or at early LLM layers. Guided by these observations, the paper proposes VScan, a two-stage visual token reduction framework. The first stage integrates global and local scanning with token merging during visual encoding, while the second stage prunes redundant visual tokens at intermediate layers of the LLM. Extensive experiments on four LVLMs and sixteen benchmarks demonstrate that VScan achieves substantial inference acceleration with minimal performance degradation.

**Strengths**
1. The paper is clear and practical, proposing VScan to balance token importance in vision–language models efficiently. The manuscript is well-structured and presents the method and results in a coherent and accessible manner.

2. Experimental validation is sufficient. The authors conduct comprehensive experiments on various tasks, demonstrating improvements to validate the effectiveness of the method. The ablation and empirical study are detailed.

3. The visualizations presented in the Appendix are impressive and highly detailed.

**Weaknesses**

1. VScan performs pruning in both the visual encoding stage and the language model stage, which raises two concerns: first, some vision encoders do not provide a [CLS] token, such as SigLIP; second, pruning during the language decoding stage makes multi-turn dialogue impossible. This contradicts the claim that the method “can be seamlessly applied to various open-sourced LVLM architectures.”

2. In Section 3, the authors use POPE and GQA as illustrative examples for analysis, but the conclusion drawn in Study 3 appears to differ from the earlier observations. I suggest adding at least two more datasets to support the conclusion and ensure its robustness.

**Audience:**

Yes

**Audience Explanation:**

In the Empirical Analysis section, the authors provide several visualizations and perspectives on visual-token pruning, which do offer some useful insights. Although these analyses are somewhat similar in spirit to FastV, the conclusions drawn here differ from those of prior work.

**Broader Impact Concerns:**

None.

**Claims And Evidence:**

No

**Claims Explanation:**

1. It appears that the reported accuracies for several baseline methods in the experiments may have been underestimated. For instance, SparseVLM's GQA accuracy under the 192 setting is reported as 57.6 in the paper, whereas the publicly available result is 59.5, which is a substantial discrepancy. Please re-check the evaluation pipeline and report the exact evaluation settings used for each baseline.

2. This paper feels like a combination of several existing ideas, and I find that it lacks a clear and distinctive motivation. Global Scan is derived from VisPruner and VisionZIP, Local Scan is from PruMerge+, token merging is based on ToMe and SparseVLM, and reducing textual irrelevance via middle-layer pruning is taken from SparseVLM. Although there are some minor differences, the overall ideas are highly similar.

**Requested Changes:**

1. In the Efficiency Analysis section, the comparison is only made between the proposed method and the original model. Please include comparisons with the other methods used in the main experiments, especially in terms of TFLOPs and latency.

---

> ### Author Response · Authors · 2025-12-03
> **Response to Reviewer k8qv (Part 1/2)**
>
> Dear Reviewer k8qv,
>
> We really appreciate your thorough review of our paper. We address the raised concerns and questions below.
>
> ---
>
> **Comment (1)**: “*VScan performs pruning in both the visual encoding stage and the language model stage, which raises two concerns: first, some vision encoders do not provide a [CLS] token, such as SigLIP; second, pruning during the language decoding stage makes multi-turn dialogue impossible. This contradicts the claim that the method “can be seamlessly applied to various open-sourced LVLM architectures*.”
>
> **Response (1)**: Thank you for your thoughtful feedback. We address your comments point by point below:
>
> - Our approach is also compatible with LVLMs that do not use a `[CLS]` token, as discussed in Section 4.1. For these models, such as Qwen-2.5-VL, we perform global token selection using self-attention scores instead. As shown in Figure 5 and Table 4, our method continues to outperform other state-of-the-art approaches when applied to Qwen-2.5-VL.
> - Adapting the middle-layer pruning component of VScan to support multi-turn conversations is straightforward: When presented with new questions, VScan can reassess token importance and reselect textually relevant visual tokens from the existing token pool through global-local scans. Although this re-selection introduces additional computation compared to text-agnostic methods (which are inherently compatible with multi-turn dialogue), the overhead is negligible relative to the overall prefill time. We have presented these remarks in Appendix A.5 for clarification.
>
> With these two clarifications, we believe our claim of “*can be seamlessly applied to various open-sourced LVLM architectures*” is valid. Please let us know if you have any remaining concerns, we are happy to further clarify or revise any ambiguous points.
>
>
> ---
>
> **Comment (2)**: “*In Section 3, the authors use POPE and GQA as illustrative examples for analysis, but the conclusion drawn in Study 3 appears to differ from the earlier observations. I suggest adding at least two more datasets to support the conclusion and ensure its robustness*.”
>
> **Response (2)**: Thanks for your valuable comments. To further support the robustness and validity of our conclusions, we additionally visualize more examples on two other benchmarks, MMBench and MME, using another model, LLaVA-NeXT-7B, as shown in Figure 4. As illustrated in the figure, convergence on the multi-choice benchmark MMBench occurs around LLM layer 20, whereas on the binary benchmark MME it occurs earlier, around layer 16. These observations are consistent with our findings on LLaVA-1.5-7B, further validating the robustness of our conclusions across both benchmarks and models.
>
> ---
>
> **Comment (3)**: “*It appears that the reported accuracies for several baseline methods in the experiments may have been underestimated. For instance, SparseVLM's GQA accuracy under the 192 setting is reported as 57.6 in the paper, whereas the publicly available result is 59.5, which is a substantial discrepancy*.”
>
> **Response (3)**: Thank you for your careful review. For convenience, we initially cited the results reported in the PyramidDrop [62] paper. Following your suggestion, we have updated the numbers to match those in the original SparseVLM paper. The revised results are provided in the revised Table 2. Please note that our approach still consistently outperforms SparseVLM, further validating the effectiveness of VScan.

---

> ### Author Response · Authors · 2025-12-03
> **Response to Reviewer k8qv (Part 2/2)**
>
> **Comment (4)**: “*This paper feels like a combination of several existing ideas, and I find that it lacks a clear and distinctive motivation. Global Scan is derived from VisPruner and VisionZIP, Local Scan is from PruMerge+, token merging is based on ToMe and SparseVLM, and reducing textual irrelevance via middle-layer pruning is taken from SparseVLM*.”
>
> **Response (4)**: We openly acknowledge that our proposed VScan leverages existing techniques in a two-stage framework. However, we wish to clarify that the goal of our work is not to introduce an entirely new method, as reflected in the wording "**rethinking**" in our title. Specifically, we systematically re-evaluate and critically analyze the design choices of existing approaches, identify key limitations, and propose effective solutions to address them. These analyses and insights are equally novel to the community, even though our work does not introduce a completely new approach.
>
> We would like to clarify, and we believe the reviewer would also agree, that our analysis is more comprehensive than any of the listed works in the context of visual token reduction. Our work provides a comprehensive analysis of how LVLMs process visual tokens during both the visual encoding and language decoding stages, highlighting key limitations of two dominant paradigms in this field. While certain aspects were partially explored by prior works, our systematic findings provide more comprehensive insights for the field of visual token reduction.
>
> Compared to existing pruning frameworks, our work provides the following unique insights:
> - Compared to **text-agnostic approaches** which typically rely on [CLS] attention in the output layer, we identify that relying solely on the output layer may overlook the rich local information encoded in the shallow layers. To address this limitation, we propose complementary local and global scans to capture more comprehensive visual details while effectively accelerating inference.
>
> - Compared to **text-aware approaches** which selectively remove tokens with low relevance to the text query during the early layers of language decoding, we identify that early layers are suboptimal for pruning due to position bias and limited engagement with visual content. To overcome this limitation, we introduce middle layer pruning to avoid position bias, preserve cross-modal interactions, and minimize the impact on final predictions.
>
> - **At a higher level**, this work introduces a novel two-stage, training-free visual token reduction framework that leverages the strengths of two lines of approaches, and enhances the efficiency of LVLMs by progressively pruning uninformative tokens during both visual encoding and language decoding stages. Our extensive experimental results across multiple benchmarks and models show that this two-stage approach significantly outperforms previous single-stage designs.
>
> ---
>
>
> **Comment (5)**: “*In the Efficiency Analysis section, the comparison is only made between the proposed method and the original model. Please include comparisons with the other methods used in the main experiments, especially in terms of TFLOPs and latency.*.”
>
> **Response (5)**: Thank you for your constructive comments. Following your suggestions, we compare the performance-efficiency trade-offs of different approaches by reporting TFLOPs, prefill time, and achieved accuracy on the POPE benchmark using LLaVA-1.5-7B under varying retention rates. The results are as follows:
>
>
> | Method | 33.3% | 22.2% | 11.1% |
> |-|:-:|:-:|:-:|
> |LLaVA-1.5-7B||3.817T / 416s / 85.9
> |FastV|1.253T / 298s / 60.7|0.836T / 266s / 57.2|0.421T / 231s / 44.5|
> |PDrop|1.253T / 307s / 82.3|0.834T / 278s / 82.3|0.415T / 240s / 55.9|
> |VisionZip|1.253T / 293s / 85.3 | 0.834T / 263s / 83.2 | 0.415T / 229s / 77.0|
> |VScan|1.253T / 301s / 86.2|0.834T / 274s / 86.1| 0.415T / 235s / 85.0|
>
> We observe that our method consistently delivers a favorable trade-off between performance and efficiency compared to previous state-of-the-arts. We have included these results and discussions into Appendix A.6 of the revised paper.
>
> ---
>
>
> We hope that our responses have addressed your concerns. If you have additional comments or concerns, please let us know and we will be more than happy to clarify further.
>
> Best,
>
> Authors

---

### Author Response · Authors · 2025-12-03
**General Response**

Dear AE and Reviewers,

We are sincerely grateful to you all for dedicating time and efforts in providing these detailed and thoughtful reviews, which helped us to improve the quality of our paper. We have also carefully revised the paper based on your thoughtful feedback. For your convenience, we have highlighted all the revisions made compared to the initial version in blue.

Here, apart from the point-by-point responses to each reviewer, we would like to summarize the contributions of this work and highlight our new results added during the rebuttal phase.

---

We are delighted that the reviewers appreciate and recognize the following strengths and contributions of this work:
- The paper is clear and practical, proposing VScan to balance token importance in vision–language models efficiently. VScan is a well-motivated and generally applicable approach to visual token compression in LVLMs. **[Reviewer k8qv, YT24]**
- This paper addresses the high computational cost of recent LVLMs, which arises from increasingly long visual token sequences produced by fine-grained visual encoding. The topic of Efficient VLMs is highly relevant to real-world applications and has gained significant attention in recent years. **[All Reviewers]**
- The methodology of VScan is well-motivated by comprehensive empirical analyses. As a researcher in this field, I have observed similar findings to those reported by the authors. **[Reviewer YT24, AvVV]**
- Extensive experiments and ablation studies demonstrate that VScan consistently outperforms recent visual compression methods for LVLMs, maintaining comparable or even superior performance to non-compressed baselines at equivalent compression ratios. **[All Reviewers]**
- The manuscript is well-structured and presents the method and results in a coherent and accessible manner.The paper is both convincing and clearly presented. **[Reviewer k8qv, AvVV]**
---

In this rebuttal, we have included the following discussions and experiments in the revised paper to address reviewers’ comments:
- In the revised Figure 4, we provide additional example studies for Study 3 using LLaVA-NeXT-7B evaluated on two additional benchmarks. These results further validate that our conclusions generalize across benchmarks and model architectures.
- In the revised Table 2, we have corrected the SparseVLM results to align with the numbers provided in their original paper.
- In the revised Appendix A.6, we present a comparison of the performance–efficiency trade-offs across methods, reporting TFLOPs, prefill time, and achieved accuracy on the POPE benchmark with LLaVA-1.5-7B under different retention rates. The results demonstrate that our approach consistently achieves a superior trade-off.
- In the revised Section 4.2, we further clarify that our VScan is compatible with both KV cache and FlashAttention.
- In the revised Study 3 in Section 3, we added explanations of how we derive our conclusions and how these conclusions inform our design, improving clarity.
- In the revised Appendix A.7, we  included the full numerical results on Qwen-2.5-VL in table format for easier future reference and comparison.
- In the revised Section 5.3, we added a more detailed discussion of the advantages and disadvantages of the text-agnostic and text-aware components in our approach.
- In the revised Tables 4 and 5, we additionally re-implement VisionZip and report its results for more comprehensive performance comparisons.

---

We greatly appreciate your time and constructive feedback on our work. If you have remaining questions or concerns, please do not hesitate to let us know and we will be happy to address them.


Best,

Authors

---

### Decision · Action_Editor_6JsE · 2026-01-08

**Recommendation:** Accept as is

**Additional Comments:**

Reviewer k8qv expressed the concern that "[the submission] feels like a combination of several existing ideas [and] lacks a clear and distinctive motivation." (k8qv) Since TMLR's acceptance criteria focus on "interest" rather than novelty or significance, that particular concern was not considered when making my official decision.

**Audience:**

Yes

**Audience Explanation:**

All reviewers agree that the submission meets the bar on the Audience criterion:

* The "visualizations and perspectives on visual-token pruning [...] offer some useful insights." (k8qv)
* "VScan is a well-motivated and generally applicable approach to visual token compression in LVLMs. [...] VScan has the potential to serve as a solid baseline for future research in this area." (YT24)
* "The topic of Efficient VLMs is highly relevant to real-world applications and has gained significant attention in recent years." (AvVV)

**Claims And Evidence:**

Yes

**Claims Explanation:**

All reviewers agree that the submission meets the bar in terms of claims and evidence:

* "The authors conduct comprehensive experiments on various tasks, demonstrating improvements to validate the effectiveness of the method. The ablation and empirical study are detailed." (k8qv)
* "The methodology of VScan is well-motivated by comprehensive empirical analyses. [...] extensive experiments and ablation studies on widely used benchmarks, consistently demonstrating its advantages over existing baselines in visual token compression for LVLMs." (YT24)
* "I find the paper both convincing and clearly presented." (AvVV)

Concerns raised in their reviews have been addressed satisfyingly by the authors' response.